# Adding value to Extended-range Forecasts in Northern Europe by Statistical Post-processing Using Stratospheric Observations

Natalia Korhonen[1,2], Otto Hyvärinen[1], Matti Kämäräinen[1], David S. Richardson[3], Heikki Järvinen[4], Hilppa Gregow[1]

[1]Weather and Climate Change Impact Research, Finnish Meteorological Institute, Helsinki, 00101, Finland
[2]Doctoral Programme in Atmospheric Sciences, University of Helsinki, Finland
[3]ECMWF, Reading, UK
[4]Institute for Atmospheric and Earth System Research/Physics, University of Helsinki, 00014, Finland

*Correspondence to*: Natalia Korhonen (Natalia.Korhonen@fmi.fi)

**Abstract.** The skill scores of the Extended-Range Forecasts (ERF) of the European Centre for Medium-Range Weather Forecasts (ECMWF) are still quite modest for the forecast weeks 3–6 in Northern Europe. As there are known stratospheric precursors impacting the surface weather with potential to improve ERFs, we aim to quantify the effect of these predictors and post-process the ERFs with them.

During boreal winter the quasi-biennial oscillation (QBO) affects the stratospheric polar vortex; the easterly (westerly) QBO often coincides with weaker (stronger) than average polar vortex. Consequently, the weaker (stronger) than average stratospheric polar vortex is connected to negative (positive) Arctic Oscillation (AO) and colder (warmer) than average surface temperatures in Northern Europe. We developed a stratospheric wind indicator, *SWI*, based on the previous weeks' stratospheric wind conditions and the phase of the AO during the following weeks. We demonstrate that there was a statistically significant difference in the observed surface temperature within the 3–6 weeks depending on the *SWI* at the start of the forecast. These temperature anomalies were underestimated by the ECMWF's reforecasts.

When our new *SWI* was applied in post-processing the ECMWF's two-week mean temperature reforecasts for weeks 3–4 and weeks 5–6 in Northern Europe during boreal winter, the skill scores of those weeks were slightly improved. This indicates there is some room to improve the ERFs, if the stratosphere-troposphere links were better captured in the modelling. In addition to this, we found that the mean skill scores of the 3–6 weeks surface temperature forecasts were higher than average in cases the polar vortex was weak at the start of the forecast.

## 1 Introduction

Extended-range forecasts (ERF; lead time up to 46 days) by dynamical models have been developed since the 1990s with the aim to fill the gap between the medium-range weather forecasts and the seasonal forecasts. It is known that ERF skills are still rather modest in forecast weeks 3–6 especially in the Northern latitudes. If the skill of the forecasts improves, ERFs have the

potential to become an essential element in climate services e.g., in the form of early warnings of climatic extremes. In an academic project CLIPS (CLImate services supporting Public mobility and Safety), climatic impact outlooks and early warnings of extremes (CLIPS forecasts) were developed by employing the ERF datasets (Ervasti et al. 2018). The CLIPS forecasts were co-designed with the general public in Finland and experimented by a one year piloting phase. As many industries, e.g., energy and food production, as well as users from the general public considered they could use and would benefit from reliable ERFs (Ervasti et al. 2018), development of more skillful ERFs is clearly needed.

The European Centre for Medium-Range Weather Forecasts (ECMWF) has produced ERFs routinely since March 2002 (Vitart 2014). The verification results of the ECMWF model's ERF (Buizza and Leutbecher 2015; Vitart 2014) on a sub-continental and a regional scale (e.g., Monhart et al. 2018) demonstrated predictive skill beyond 2 weeks for temperature reforecasts over Northern Europe. ECMWF uses bias-correction of the mean in their automatic products, removing the mean bias computed from the reforecasts, depending on the time of the year (Buizza and Leutbecher 2015). We consider the bias over Northern Europe not to be dependent only on the time of the year but also on the prevailing weather pattern, and therefore, we aim to explore whether known teleconnections such as the strength of the stratospheric polar vortex, the phase of the Arctic Oscillation (AO), and the phase of the Quasi-Biennial Oscillation (QBO) could be used in improving the forecasts.

The stratospheric polar vortex is an upper-level low-pressure area that forms over both the northern and southern poles during winter due to the growing temperature gradient between the pole and the tropics. Strong westerly winds circulate the polar vortex, isolating the gradually cooling polar cap air. The strength of the northern polar vortex varies from year to year and can be indicated by, e.g., the zonal mean zonal wind (ZMZW) at 60°N and 10 hPa or polar cap temperatures. The stronger the circumpolar winds and the colder the polar cap temperatures are, the stronger is the polar vortex. Planetary waves from the troposphere disturb the northern stratospheric polar vortex, leading to meandering and weakening of the westerlies and occasionally to reverse, i.e., easterly flow (Schoeberl, 1978). This weakening of the stratospheric polar vortex also leads to warming of the polar cap temperatures, sometimes even > 30–40 K within several days. A warming of this magnitude together with a reversal of the ZMZW at 60°N and 10 hPa is commonly defined as a major sudden stratospheric warming (SSW), albeit other definitions have also been used (Butler et al. 2015).

During boreal winter the strength of the polar vortex affects the phase of the AO, which characterizes air mass flow between the Arctic and the mid-latitudes. At the surface, the AO index is affected by the strength of the polar vortex with a time lag of about two to three weeks (Baldwin and Dunkerton 1999). A strong polar vortex is characterized by lower than average surface pressure in the Arctic, positive AO index, and strong westerly winds keeping the cold Arctic air locked in the polar region and bringing milder and wetter than average weather to Northern Europe (Limpasuvan et al. 2005). In contrast, a weak polar vortex is characterized by higher than average surface pressure in the Arctic, negative AO index, and the meandering and/or

weakening of the polar jet stream and tropospheric jet stream enabling cold arctic/polar air outbreaks to Northern Europe (Thompson et al. 2002, Tomassini et al. 2012).

During boreal winters, the strength of the stratospheric polar vortex influences the surface weather in the Northern Hemisphere within weeks or months (Baldwin and Dunkerton 2001, Kidston et al. 2015), hence holding a potential for forecasting in that time scale. When making forecasts based on the strength of the polar vortex, a noteworthy phenomenon is also the QBO, a quasiperiodic oscillation of the equatorial zonal wind between downwards propagating easterlies and westerlies in the tropical stratosphere with a mean period of 28 to 29 months (Baldwin et al. 2001). Holton and Tan (1980) found that during the easterly QBO at level 50 hPa the polar vortex was statistically significantly weaker than during westerly QBO at the same level. Further, Scaife et al. (2014) demonstrated indicators of a more negative AO in the easterly QBO at level 30 hPa than in the westerly QBO phase at this level. There is no precise consensus of the mechanisms of this tropical-extratropical connection (Garfinkel et al. 2012), but the most common explanation is that the QBO affects the polar vortex via the Holton Tan effect: During easterly QBO, small amplitude planetary waves are reflected back towards the North Pole weakening the polar vortex (Holton and Tan 1980, 1982; Watson and Gray 2014, Gray et al. 2018).

Challenges related to the realistic modelling of the dynamical stratosphere-troposphere coupling have been adduced by Shepherd et al. (2018) and Polichtchouk et al. (2018). In the ECMWF model the skill of the QBO forecasts decreases substantially after the first 2 months of the forecast. This is shown to be sensitive to the parametrization of the tropical non-orographic gravity wave drag in the model (Polichtchouk et al. (2018), Polichtchouk et al. (2017)). The amplitude of the QBO tends to weaken (in the ECMWF model) through the forecast. Even though some S2S models, including the ECMWF's Integrated Forecasting System (IFS, Vitart, 2014), are already able to reproduce the QBO's effect on the polar vortex, they are still underestimating the effect on surface weather (Garfinkel et al. 2018). Furthermore, the anomalous QBO disruption in winter 2015/2016 was not forecasted by the models (Newman et al. 2016).

In this paper, we first verify the raw and the mean bias-corrected surface temperature reforecasts of the ECMWF's ERFs for forecast weeks 1 to 6 over Northern Europe against ERA-Interim surface temperature re-analysis (Dee et al. 2011). After that, our aim is to find out which stratospheric observations available at the start of the forecast are followed by a statistically significantly weaker AO index. For this, we explore the observed daily AO index during boreal winters 1981–2016, 1–2 weeks, 3–4 weeks, and 5–6 weeks after different phases of QBO and strengths of the observed stratospheric zonal mean winds. According to the observed daily AO index, after different phases of QBO, and strengths of the observed stratospheric zonal mean winds, we define a Stratospheric Wind Indicator (*SWI*), which is a novel indicator for the strength of the AO index in the following 1 to 6 weeks. For a statistically significantly weaker mean AO index, the *SWI* is defined as $SWI_{neg}$; otherwise, *SWI* is defined as $SWI_{plain}$. Further, we study the mean surface temperature anomalies observed in Northern Europe 1–2 weeks, 3–4 weeks, and 5–6 weeks after strong or weak ZMZW at 60°N and 10hPa and after $SWI_{neg}$ versus $SWI_{plain}$. We utilize $SWI_{neg}$

and $SWI_{plain}$ in post-processing the mean of the temperature forecasts of ECMWF reforecasts. Finally, we compare the $SWI$ based post-processed ECMWF reforecasts with the mean bias-corrected ECMWF reforecasts. Our paper is constructed as follows: First, we present the datasets and methods. Then, we present results about the selection of the $SWI_{neg}$ and $SWI_{plain}$ and the skill scores of the forecasts without post-processing and with post-processing. In the Discussions and Conclusions section,

we present our view on our findings and the possible next steps.

## 2 Datasets and Methods

We verified and post-processed ERFs of the ECMWF's IFS Cycle 43r1 (Vitart, 2014), which belongs to the models of the Sub-seasonal to seasonal (S2S) prediction project of the World Weather Research Program/World Climate Research Program (Vitart et al. 2017). These forecasts are run twice a week, on Mondays and Thursdays, in a horizontal resolution of 0.4 degrees.

We first studied the weekly mean temperatures of the Monday runs over Northern Europe (52° N to 71.2° N and 10° E to 33.2° E) with lead times of 1 to 6 weeks, here called forecast weeks 1 to 6. We verified the 20 years × 52 weeks = 1040 reforecasts (11 members ensemble) for 1997–2016 run for the same dates as the operational forecasts, i.e., as Mondays in 2017. The weekly averages of the raw, mean bias-corrected (Section 2.2), and post-processed (Sections 2.3 and 2.4) surface temperature forecasts over Northern Europe were verified against ERA-Interim 1981–2016 temperature re-analyses (Dee et al. 2011). Years

1981–2010 of the ERA-Interim data were used as the climatological reference period and as the statistical/climatological forecast.

### 2.1 Skill scores of the forecasts

A commonly used measure for the probabilistic forecasts is the continuous ranked probability score (CRPS, Hersbach 2000) calculated by the following Eq. (1):

$$CRPS = \int |F(y) - F_o(y)|^2 dx, \tag{1}$$

where $F(y)$ and $F_o(y)$ are the cumulative distribution functions of the forecast and the observation, respectively.

The CRPSs were calculated by the R package 'ScoringRules' (Jordan et al. 2018) for the ECMWF's reforecast ($CRPS_{rf}$) and the climatological forecasts (ERA-Interim weekly mean temperatures in 1981–2010), which were used as the reference ($CRPS_{clim}$). As the ensemble size of the reforecasts, $m$, was only 11, and the ensemble size of the operational forecasts of the

ECMWF's IFS, $M$, was 51, the expected CRPS, the $CRPS_{RF}$ of the ECMWF's reforecast was calculated for 51 members using equation 26 in Ferro et al. (2008):

$$CRPS_{RF} = \frac{m(M+1)}{M(m+1)} CRPS_{rf} \tag{2}.$$

We calculated the annual means of the expected CRPS$_{RF}$ across all weeks (1 to 52) of the years 1997–2016 reforecasts. These annual means were computed separately for lead times of 1 week, 2 weeks, 3 weeks, 4 weeks, 5 weeks, and 6 weeks, here

called forecast week 1, forecast week 2, forecast week 3, forecast week 4, forecast week 5, and forecast week 6, respectively. Further, the skill scores of the annual mean CPRSs, the annual mean CRPSSs, for each lead time were calculated as follows:

$$CRPSS = 1 - \frac{CRPS_{RF}}{CRPS_{clim}}$$ (3).

The statistical significances of each forecast week's annual mean CRPSS was determined for each grid point. The p-value with the null hypothesis that the CRPSS is zero was calculated by bootstrap resampling procedure with replacement and a sample size of 5000 for significance level 0.05.

## 2.2 Bias-correction of the ensemble mean

The mean bias-correction (as in Buizza and Leutbecher 2015, eq. 7a) removed the mean bias computed from the ensemble reforecasts for the 20 years (1997–2016) depending on the forecast week date. For the 1997–2016 reforecasts, the average bias was calculated considering $19 \times 11 \times 5 = 1045$ ensemble reforecast members: 11 members' reforecast with initial dates defined by five weeks centred on the forecast week date for the 19 years reforecasts (1997–2016 excluding the reforecast year). The mean bias-corrected weekly mean temperatures were verified against the ERA-Interim data by calculating the annual mean CRPS separately for each lead time, i.e., forecast weeks 1 to 6. The skill scores of the mean bias-corrected forecasts and their statistical significance were calculated as explained in Section 2.1.

## 2.3 Definition of the stratospheric wind indicator (*SWI*)

As numerous observational and modelling studies have shown, the stratospheric polar vortex influences the weather in the Northern Hemisphere during boreal winter; strong polar vortex coincides more often with positive AO index and mild surface weather in Northern Europe, whereas weak polar vortex is more often followed by negative AO index and cold air outbreaks (Thompson and Wallace 1998, 2001, Kidston et al. 2015 and references therein). We aimed to find stratospheric precursors for a statistically significantly weaker AO index available at the start of the forecast. The daily surface AO index were downloaded from the National Centers for Environmental Prediction (NCEP), Climate Prediction Center (CPC). This daily AO index from the NCEP CPC is produced by projecting the daily 1000 hPa geopotential height anomalies north of 20° N onto the loading pattern of AO, which is defined as the first leading mode from the Empirical Orthogonal Function (EOF) analysis of monthly mean 1000 hPa height anomalies poleward of 20° N during 1979–2000. As precursors for the AO index we used two stratospheric wind data sets. The first data were the daily ZMZW at 60°N and 10 hPa during 1981–2016 of the Modern-Era Retrospective analysis for Research and Applications, Version 2 (MERRA-2; Rienecker et al. 2011) reanalysis data provided by the National Aeronautics and Space Administration (NASA). The second data were the monthly mean zonal wind components at levels 70 hPa, 50 hPa, 40 hPa, 30 hPa, 20 hPa, 15 hPa, and 10 hPa from the Singapore radio soundings, during 1981–2016, provided by the Free University of Berlin, representing the equatorial stratospheric monthly mean zonal wind components, the QBO (Naujokat 1986).

We explored the mean AO index after the beginning of each week in November–February (1981–2016). First, we investigated the mean AO index 1–2, 3–4, and 5–6 weeks after the minimum daily ZMZW at 60°N and 10 hPa during the preceding 10 days had been:

- below its overall wintertime (November–March 1981–2016) 10th percentile, corresponding a value of 3.8 ms$^{-1}$, indicating a weak polar vortex,

- between its overall wintertime (November–March 1981–2016) 10th and 20th percentile corresponding to values greater than 3.8 ms$^{-1}$ and lower than 12 ms$^{-1}$,

- between its overall wintertime (November–March 1981–2016) 20th and 50th percentile corresponding to values greater than 12 ms$^{-1}$ and lower than 27 ms$^{-1}$,

- between its overall wintertime (November–March 1981–2016) 50th and 80th percentile corresponding to values greater than 27 ms$^{-1}$ and lower than 41 ms$^{-1}$, and

- above its overall wintertime (November–March 1981–2016) 80th percentile corresponding to values greater than 41 ms$^{-1}$, indicating a strong polar vortex.

In the cases the ZMZW at 60°N and 10 hPa was between 3.8 ms$^{-1}$ and 41 ms$^{-1}$ during the preceding 10 days, we explored the mean AO index 1–2, 3–4, and 5–6 weeks after following predictors:

- westerly QBO at 30 hPa, the *WQBO*,

- easterly QBO at 30 hPa, the *EQBO,*

- *EQBO* with the maximum of the monthly mean zonal wind components of the QBO between 70 hPa and 10hPa restricted to 7ms$^{-1}$, 10ms$^{-1}$, and 13ms$^{-1}$,

The statistical significance of the difference between the AO index following two different stratospheric situations, e.g., the *EQBO* and the *WQBO*, was determined using a two-sided Student's t-test with the null hypothesis that there is no difference. The statistically significant predictors for weaker AO index observed 1–2 weeks, 3–4 weeks, and 5–6 weeks after these stratospheric situations, were used to define a *SWI* to be *SWI$_{neg}$*; otherwise, it was defined as *SWI$_{plain}$* for the beginning of each winter week (November–February) in 1981–2016.

**2.4 Utilizing the stratospheric winds indicator (*SWI*) in forecasting**

In this section, we investigated the observed and reforecasted surface temperature anomalies 1–2 weeks, 3–4 weeks, and 5–6 weeks after weak/strong ZMZW at 60°N and 10 hPa and *SWI$_{neg}$*/*SWI$_{plain}$* defined in Section 2.3. First, we calculated the two-week mean temperature anomalies of the ERA-Interim reanalyses (Dee et al. 2011) of the 1–2 weeks, the 3–4 weeks, and the 5–6 weeks from the beginning of each week in January, February, November, and December in 1981–2016 in Northern Europe. Subsequently, we divided the observed two-week mean temperature anomalies to sets of anomalies, representing weak ZMZW at 60°N and 10 hPa (below 3.8 ms$^{-1}$), strong ZMZW at 60°N and 10 hPa (above 41 ms$^{-1}$), *SWI$_{neg}$*, and *SWI$_{plain}$* according to the

previous weeks' stratospheric wind conditions. Thereafter, we determined the statistical significance of the difference between the surface temperatures after $SWI_{neg}$ and $SWI_{plain}$ (and after weak/strong ZMZW at 60°N and 10 hPa in comparison to the rest of the cases) using a two-sided Student's t-test with the null hypothesis that there is no difference between $SWI_{neg}$ and $SWI_{plain}$ (and no difference between weak/strong ZMZW at 60°N and 10 hPa and the rest of the cases). This same procedure to define

5 the difference between the surface temperatures after $SWI_{neg}$ and $SWI_{plain}$ (and weak and strong ZMZW at 60°N and 10 hPa) was used for the ERA-Interim reanalyses for the period 1997–2016 to see how the selection of a shorter period affects the temperature anomalies. Further, the mean surface temperature anomalies 1–2 weeks, 3–4 weeks, and 5–6 weeks after $SWI_{neg}$ and $SWI_{plain}$ (and after weak and strong ZMZW at 60°N and 10 hPa) in the ECMWF reforecasts run in the beginning of each week in November–February 1997–2016 were calculated to examine how the model reproduced the anomalies.

For post-processing the ECMWF reforecasts in cases the ZMZW at 60°N and 10 hPa was between 3.8 ms$^{-1}$ and 41ms$^{-1}$ at the start of the forecast, we calculated $TA_{SWIneg}$ and $TA_{SWIplain}$ representing mean temperature anomalies in November–February 1981-2016 after $SWI_{neg}$ and $SWI_{plain}$, respectively. The means, $TA_{SWIneg}$ and $TA_{SWIplain}$ representing $SWI_{neg}$ and $SWI_{plain}$ were calculated separately for each 0.4° × 0.4° grid point over Northern Europe.

For the post-processing of the ECMWF reforecasts, we first defined the $SWI$ either $SWI_{neg}$ or $SWI_{plain}$ at the start of the forecast according to previous weeks' stratospheric wind conditions. According to the $SWI$, we added either $TA_{SWIneg}$ or $TA_{SWIplain}$ to the ERA-Interim mean temperature during 1981–2016, corresponding to forecast weeks 1–2, 3–4, and 5–6 to get a $SWI_{neg}$ and $SWI_{plain}$ based mean temperatures, $T_{SWIneg}$ and $T_{SWIplain}$, for weeks 1–2, 3–4, and 5–6, respectively. The $T_{SWIneg}$ and $T_{SWIplain}$ were

20 used in post-processing the ECMWF reforecasts' mean bias-corrected ensemble members, $T_{BC}$, by calculating a weighted average, $T_{SWI\_BC}$, for $SWI_{neg}$ as follows:

$$T_{SWI\_BC} = (1 - k_{SWI}) * T_{BC} + k_{SWI} * T_{SWIneg} \tag{4}$$

And for $SWI_{plain}$,

$$T_{SWI\_BC} = (1 - k_{SWI}) * T_{BC} + k_{SWI} * T_{SWIplain} \tag{5}$$

25 where $T_{SWI\_BC}$ was a post-processed ensemble member. $k_{SWI}$ was the weight of the $T_{SWIneg}$ or $T_{SWIplain}$, which was tested between 0–1 and defined according to the best improvement in the skill scores of the post-processed forecast. By Eq. (4) and Eq. (5), we adjusted each ensemble member with the same weight, and hence, the original spread of the ECMWF reforecasts remained unchanged. The skill scores of the $SWI$ based post-processed forecasts, and their statistical significance, were calculated as explained in Section 2.1.

## 3 Results

### 3.1 Skill scores of the forecasts

The annual mean of the expected CRPSS and its 95% level of confidence of the raw and the mean bias-corrected (Section 2.2) weekly mean temperature of the ECMWF reforecasts for 1997–2016 are displayed in Figure 1. In grid points where the CRPSS was higher than zero and the confidence level was higher than 95% (dotted areas), the reforecasts were statistically significantly better than just the statistical forecast based on 1981–2010 climatology. Figure 1 illustrates that for forecast weeks 1–6 the mean bias-corrected ERF reforecasts were on average significantly better than climatology. The annual mean CRPSS values show that in forecast weeks 1–3 the CRPSSs are for the most part above 0.1, whereas on in forecast weeks 4–6 they are mostly lower, between 0 and 0.1.

### 3.2 The stratospheric observations and the thereafter observed AO index and surface temperature

Figure 2 and Fig. 3 show boxplots of the observed mean of the daily AO index 1–2 weeks, 3–4 weeks, and 5–6 weeks after different strengths of the ZMZW at 60°N and 10 hPa (Fig. 2) and after different phases of QBO and restrictions in the strength of the stratospheric winds in 1981–2016 (Fig. 3). In Fig. 2 the first box (brown) represents the mean AO index after all the cases in 1981–2016 November–February, i.e., 36 years * 17 weeks=612 cases. The 2nd box (blue) in Fig. 2 shows the mean AO index after cases where the daily ZMZW at 60°N and 10 hPa was below its 10th percentile (3.8ms$^{-1}$) during the preceding 10 days, corresponding to cases with a weak polar vortex. The observed mean AO index was statistically significantly weaker at the 99% confidence level 1–2 weeks, 3–4 weeks, and 5–6 weeks after the daily ZMZW at 60°N and 10 hPa had been below its overall wintertime 10th percentile (indicating a weak polar vortex). The third, fourth, fifth, and sixth (green, yellow and purple, red) boxes in Fig. 2 represent the mean AO index after cases the daily ZMZW at 60°N and 10 hPa was between its 10$^{th}$ and 20$^{th}$ percentile (between 4 ms$^{-1}$ and 12 ms$^{-1}$), 20$^{th}$ and 50$^{th}$ percentile (between 12 and 27 ms$^{-1}$), 50$^{th}$ and 80$^{th}$ percentile (between 27 ms$^{-1}$ and 41 ms$^{-1}$), and above 41 ms$^{-1}$, respectively. In these cases the AO index seems to be the statistically significantly higher 1–2 weeks (Fig. 2a) after the ZMZW at 60°N and 10 hPa was between 27 ms$^{-1}$ and 41 ms$^{-1}$ and above 41 ms$^{-1}$.

In Fig. 3 the first box (light brown) represents the mean AO index after all the cases in 1981–2016 November–February, with ZMZW at 60°N and 10 hPa between 3.8 ms$^{-1}$ and 41 ms$^{-1}$. The second and third boxes show the mean AO index after easterly (*EQBO*, blue) and westerly (*WQBO*, pink) QBO at the 30 hPa level, respectively, in cases the ZMZW at 60°N and 10 hPa was between 3.8 ms$^{-1}$ and 41 ms$^{-1}$. The p-value written below each boxplot pair indicates the likelihood of such a pair of distributions arising from a random sampling of a single distribution as given by a Student's t-test, i.e., p-values less than 0.05 indicate that the means of the data sets differ significantly at the 95% level of confidence. The median and the mean of the mean AO index 5–6 weeks after *EQBO* were statistically significantly lower than after *WQBO*. The *EQBO* (blue) box shows all the cases of *EQBO* with no restriction in the QBO's monthly mean zonal wind components, whereas the fourth, the sixth,

and the eighth blueish boxes show the mean AO index after *EQBO* with all the QBO's monthly mean zonal wind components between levels 70…10hPa being below 13 ms$^{-1}$, 10 ms$^{-1}$, and 7 ms$^{-1}$, respectively. Restricting the *EQBO* cases by a maximum of the QBO's monthly mean zonal wind components in levels 70…10hPa being below 13 ms$^{-1}$ or 10 ms$^{-1}$ led to a statistically significantly lower  mean AO during the following 1–2, 3–4, and 5–6 weeks in comparison to the rest of the cases.

Aiming to select stratospheric precursors indicating weak AO index, we defined the *SWI* to be negative in cases when the QBO was easterly at 30 hPa and the QBO's monthly mean zonal wind components in levels 70…10hPa were weaker than 10ms$^{-1}$ and the minimum of the daily ZMZW at 60°N and 10 hPa during the 10 last days was between its overall wintertime 10th and 80$^{th}$ percentile, i.e., above 3.8 ms$^{-1}$ and below 41 ms$^{-1}$. In other cases, when the minimum of the daily ZMZW at 60°N and 10

hPa during the 10 last days month was above 3.8 ms$^{-1}$ and below 41 ms$^{-1}$, the *SWI* was defined as plain. This decision tree for the *SWI* is depicted in Fig. 4.

Figure 5 shows the ERA-Interim (1981–2016 and 1997–2016) and model forecasted mean temperature anomalies 1–6 weeks after weak (below 3.8 ms$^{-1}$) and strong (higher than 41 ms$^{-1}$) ZMZW at 60°N and 10 hPa. Cases with ZMZW at 60°N and 10

hPa weaker than 3.8 ms$^{-1}$, in Fig. 5a–c and Fig 5g–i (stronger than 41 ms$^{-1}$ in Fig. 5d–f and Fig. 5j–l) were followed by colder (warmer) than average mean temperature. The reforecasts (Fig. 5 m-r) capture these anomalies clearly, in some areas even too strong in comparison to the ERA-Interim 1997-2016.

Figure 6 shows the ERA-Interim (periods 1981–2016 and 1997–2016) and model forecasted (the period 1997–2016) mean

temperature anomalies of the weeks' 1–2, 3–4, and 5–6 in November–February after *SWI$_{neg}$* and *SWI$_{plain}$*. The ERA-Interim showed on average lower mean temperatures for the weeks' 1–2, 3–4 and 5–6 after *SWI$_{neg}$* (Fig. 6a-c and 6g-i). The ECMWF reforecasts also showed cold anomalies after *SWI$_{neg}$* for forecast weeks 1–2 (Fig. 6m)  but  weaker than the observed ones. For forecast weeks 3–6 (Fig. 6n-o) there was no sign of cold anomalies in the mean of the reforecast. Further, the ERA-Interim reanalyses showed on average higher mean temperatures for weeks 1–2, 3–4, and 5–6 after *SWI$_{plain}$* (Fig. 6d–f and 6j–l). This

warm anomaly was well forecasted in the forecasts weeks 1–2 (Fig. 6p), and it was partly but weaker forecasted in forecast weeks 3–4 (Fig. 6q) and 5–6 (Fig. 6r). The mean temperature anomalies 1–6 weeks after *SWI$_{neg}$* (Fig. 6a–c) and *SWI$_{plain}$* (6d–f) during 1981–2016 were statistically significantly different using a Student's t-test, with anomalously cold surface temperatures more common 1-6 weeks after *SWI$_{neg}$*. When examining the years 1997–2016 (Fig. 6g–i and Fig. 6j–l), which was the reforecast period, the temperature anomalies were mostly of the same sign than during the longer 1981–2016 period

(Fig. 6a–c and Fig. 6d–f).

### 3.3 The *SWI* and the forecasted mean temperatures

The mean temperature anomalies in Fig. 6(a–f) for Northern Europe were used for the *SWI* based post-processing as described in Section 2.4. The CRPSS of the mean temperature of the forecast weeks 1–2 were not improved by the *SWI* (no figure),

whereas the CRPSSs of the mean temperatures of the forecast weeks 3–4 and 5–6 were improved by the *SWI* based post-processing (Fig. 7a and 7b). The best median CRPSS was achieved by $k_{SWI}$=0.4, for forecast weeks 3–4 and by $k_{SWI}$=0.7 for forecast weeks 5–6.

Figure 8a–f shows the forecast skill of the mean bias-corrected mean temperature reforecasts of forecast weeks 3–4 and 5–6 in cases the ZMZW at 60°N and 10 hPa was weak (below 3.8 ms$^{-1}$, Fig. 8a–b), strong (above 41 ms$^{-1}$, Fig. 8c–d), and between 3.8 ms$^{-1}$ and 41ms$^{-1}$ (Fig. 8e–f) at the start of the forecast. In cases of the weak ZMZW at 60°N and 10 hPa at the start of the forecast (Fig. 8a–b) the CRPSSs reached even higher than 0.4 values in some areas, indicating higher predictability than cases with ZMZW at 60°N and 10 hPa stronger than 3.8 ms$^{-1}$ (Fig. 8c–f). Also in cases the ZMZW at 60°N and 10 hPa was strong

(higher than 41 ms$^{-1}$) at the start of the forecast, in forecast weeks 5-6 the mean CRPSS reached in some areas 0.2 (Fig. 8d), showing higher predictability than in cases the ZMZW at 60°N and 10 hPa was between 3.8 ms$^{-1}$ and 41 ms$^{-1}$ (see Fig. 8f).

Figures 8e–f show the forecasts skill of the mean temperature of the forecast weeks 3–4 and weeks 5–6 forecasted by the mean bias-corrected reforecasts alone in cases the ZMZW at 60°N and 10 hPa was between 3.8 ms$^{-1}$ and 41 ms$^{-1}$ at the start of the

forecast. Figure 8g and Fig. 8h depict  the mean bias-corrected and *SWI* post-processed reforecasts in cases the ZMZW at 60°N and 10 hPa was between 3.8 ms$^{-1}$ and 41 ms$^{-1}$ at the start of the forecast ($k_{SWI}$ =0.4 and $k_{SWI}$ =0.7 in  Fig. 8g and Fig 8h, respectively). By using the *SWI* based post-processing to the ECMWF forecasts, the CRPSSs for weeks 3–4 and weeks 5–6 were slightly improved and the area of these forecasts being significantly better than just the climatological forecast was expanded.

**4 Discussion and Conclusions**

Based on ECMWF's extended-range reforecasts for the period 1997–2016, we found that the weekly mean surface temperature forecasts over Northern Europe were on average significantly better than just the climatological forecast in weeks 1–6, however, in weeks 4–6, the CRPSSs were quite low, mostly between 0 and 0.1.

We studied the mean AO index after different thresholds of ZMZW at 60°N and 10 hPa. We found that the mean AO index was statistically significantly weaker 1–2 weeks, 3–4 weeks, and 5–6 weeks after the daily ZMZW at 60°N and 10 hPa had been below their overall wintertime 10th percentile, 3.8 ms$^{-1}$ (indicating a weak polar vortex).

We separated the weak (below 3.8 ms$^{-1}$) and strong (above 41 ms$^{-1}$) ZMZW at 60°N and 10 hPa cases and investigated the rest

of the data based on the QBO. We showed that in addition to the previously demonstrated more negative AO during easterly QBO in comparison to westerly QBO at 30 hPa (Scaife et al. 2014), the mean AO index was sensitive to the maximum strength of the QBO's monthly mean zonal wind components in levels 70…10hPa during the easterly QBO at 30 hPa. Based on

observations, we found that the mean AO index was statistically significantly weaker 1–2 weeks, 3–4 weeks, and 5–6 weeks after the monthly mean QBO was easterly at 30 hPa, and all the QBO's monthly mean zonal wind components in levels 70…10hPa were less than 10 ms$^{-1}$. Selecting the $SWI_{neg}$ to include cases where the QBO was easterly at 30 hPa and all the QBO's monthly mean zonal wind components in levels 70…10hPa were less than 10 ms$^{-1}$ resulted in a statistically significantly weaker AO index within the following 1–6 weeks in comparison to the rest of the data, defined as $SWI_{plain}$. As negative AO index enables cold air outbreaks to Northern Europe (Thompson et al. 2002, Tomassini et al. 2012) and positive AO index tends to bring milder and wetter than average weather to Northern Europe (Limpasuvan et al. 2005), we investigated how the mean surface temperatures were in November-February (1981–2016) in Northern Europe 1–6 weeks after weak/strong ZMZW at 60°N and 10 hPa and after $SWI_{neg}$/$SWI_{plain}$. We found that the mean surface temperature anomalies in Northern Europe in November–February in 1981–2016 after weak and strong ZMZW at 60°N and 10 hPa ($SWI_{neg}$ and $SWI_{plain}$) were mostly statistically significantly different, with anomalously cold surface temperatures more common 1–6 weeks after weak ZMZW at 60°N and 10 hPa ($SWI_{neg}$). The mean temperature anomalies corresponding to $SWI_{neg}$/$SWI_{plain}$ were used in post processing the ECMWF's mean temperature reforecast for weeks 3–4 and 5–6 in Northern Europe during boreal winter, and thereby, those weeks' forecast skills were slightly improved.

We also investigated the forecast skill in cases of strong or weak ZMZW at 60°N and 10 hPa. We found that weak ZMZW at 60°N and 10 hPa at the start of the forecast led to higher than average predictability of surface temperature anomalies for forecast weeks 3–6. This could be valuable information for both forecasters and customers and should be further researched.

This study demonstrates that the QBO-polar vortex connection should be better integrated into the extended-range surface temperature forecasts over Northern Europe. The $SWI$ based post-processing method introduced in this paper could also be tested for other northern areas affected by the polar vortex and to precipitation and windiness forecasts, and it could be further developed by, e.g., the Madden-Julian-Oscillation (Madden and Julian 1994; Zhang 2005; Jiang et al. 2017; Vitart 2017; Vitart and Molteni 2010; Robertson et al. 2018, Cassou 2008). In this study, the effect of global warming was not filtered from the temperature anomalies used for statistical post-processing. In future work, the impact of filtering the effect of global warming could be tested. Moreover, the next step would be looking for the stratospheric signal from the forecast model.

*Data availability*. ERA-Interim data available at https://apps.ecmwf.int/datasets/data/interim-full-daily/levtype=sfc/ (last accessed 24 June 2019). ECMWF reforecasts data available at https://apps.ecmwf.int/mars-catalogue/ (last accessed 28 June 2019). AO index data available at https://www.cpc.ncep.noaa.gov/products/precip/CWlink/daily_ao_index/ao.shtml (last accessed 24 June 2019). The daily ZMZW at 60°N and 10 hPa data available at https://acd-ext.gsfc.nasa.gov/Data_services/met/ann_data.html (last accessed 24 June 2019). The QBO data data available at https://www.geo.fu-berlin.de/met/ag/strat/produkte/qbo/qbo.dat (last accessed 24 June 2019). The data of Figures 1–3 and 5–8 available at https://etsin.fairdata.fi/dataset/9f35af27-c3bb-4115-9e31-a9ef339b10ed.

*Competing interests.* The authors declare that they have no conflict of interest.

*Author contributions.* NK designed the study, analysed the results and prepared the manuscript with contributions from all co-authors. OH participated in the study design and analysing the results. MK contributed to the discussions and fine-tuned the experiments. DSR contributed to the discussions and to the interpretation of the results. HJ provided supervision during the experiments and writing. HG contributed to the study design and was in charge of the management and the acquisition of the financial support for the CLIPS-project leading to this publication.

*Acknowledgements.* We wish to thank Academy of Finland for funding the project (number 303951 SA CLIPS). We also acknowledge the ECMWF for monthly forecast data and ERA-Interim data, NOAA/CPC for providing the AO index data, NASA for providing 10hPa wind data, and Free University of Berlin for providing the QBO data. We thank the CLIPS team and developers of the R cran calculation package 'ScoringRules'. We thank the three anonymous reviewers for their good and constructive comments.

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

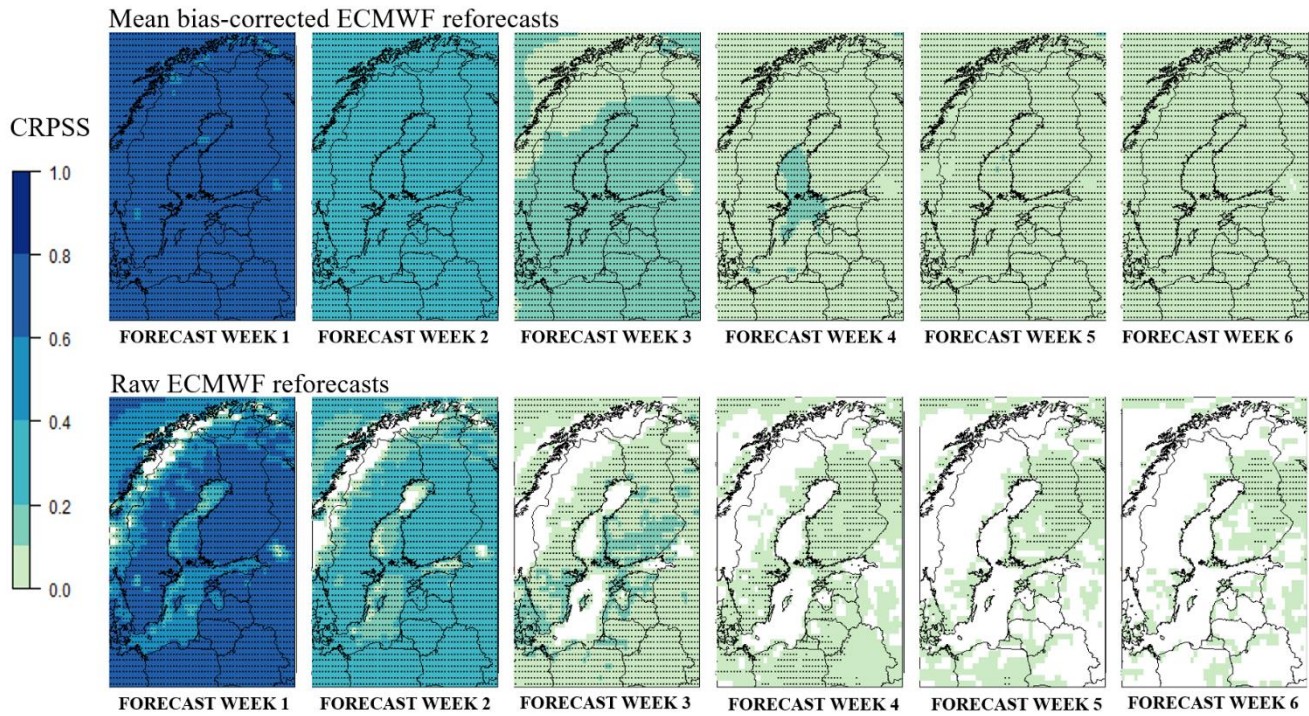

**Figure 1: Annual mean of the expected CRPSS of the weekly mean temperature of the mean bias-corrected (upper row) and raw (lower row) ECMWF reforecasts for years 1997–2016 using ERA-Interim climatology of 1981–2010 as the reference. The dotted areas represent the 95% level of confidence that the CRPSS is above zero.**

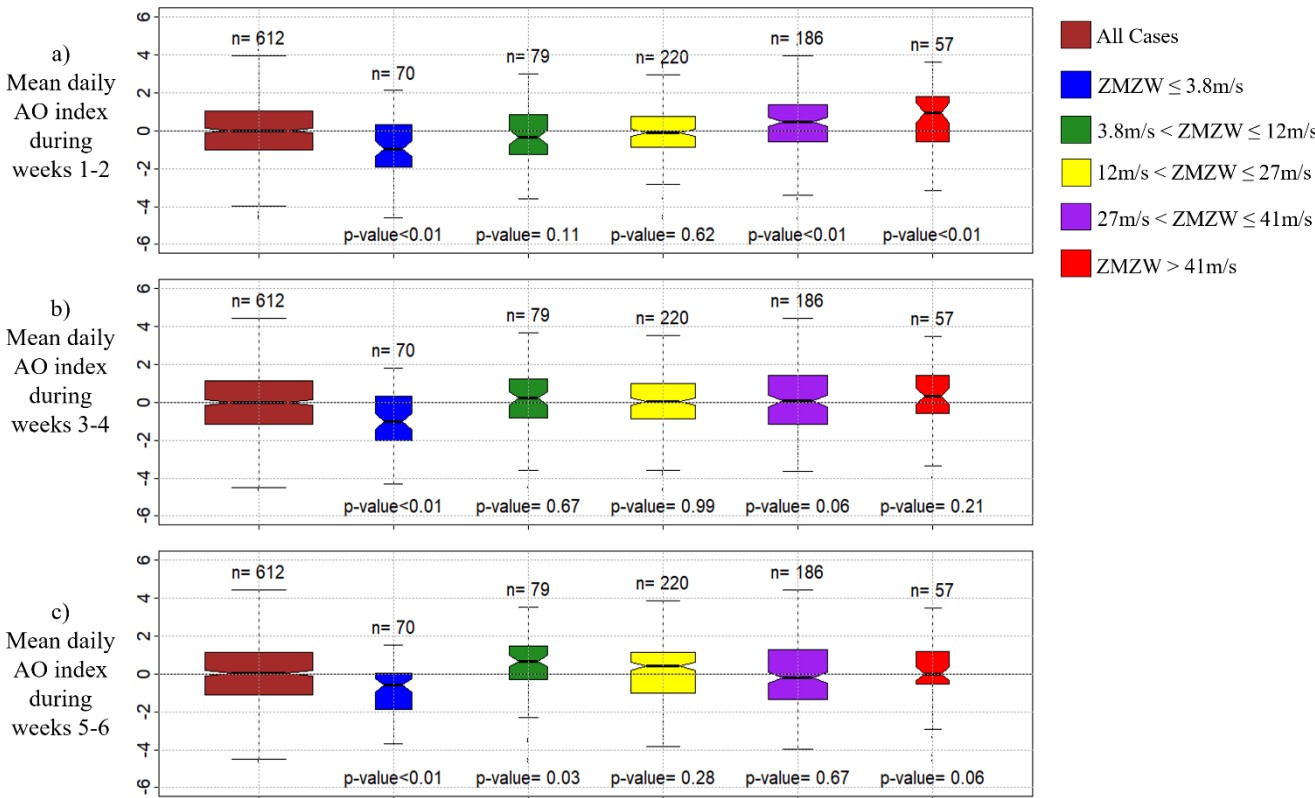

**Mean AO index after different thresholds of the Zonal Mean Zonal Wind at 60°N and 10 hPa (ZMZW)**

**Figure 2** Observed mean AO index in November-March (1981-2016) a) 1–2, b) 3–4 and c) 5–6 weeks after different thresholds of the zonal mean zonal wind at 60°N and 10 hPa (ZMZW). The horizontal line dividing each box into two parts shows the median of the data, the ends of the box show the lower and upper quartiles, and the whiskers represent the highest and the lowest values excluding outliers. The n written above each box indicates the number of observations in each group. The widths of the boxes have been drawn proportional to the square-roots of n. The p-value written below each boxplot pair indicates the likelihood of such a pair of distributions arising from a random sampling of a single distribution as given by a Student's t-test, i.e., p-values less than 0.05 indicate that the means of the data sets differ significantly at the 95% level of confidence. The notches of each side of the boxes were calculated by R boxplot.stats. If the notches of two plots do not overlap, this is 'strong evidence' that the two medians differ (Chambers et al., 1983, p. 62).

**Mean AO index after easterly QBO (*EQBO*) versus westerly QBO (*WQBO*) at 30 hPa in cases ZMZW between 3.8 m/s and 41 m/s**

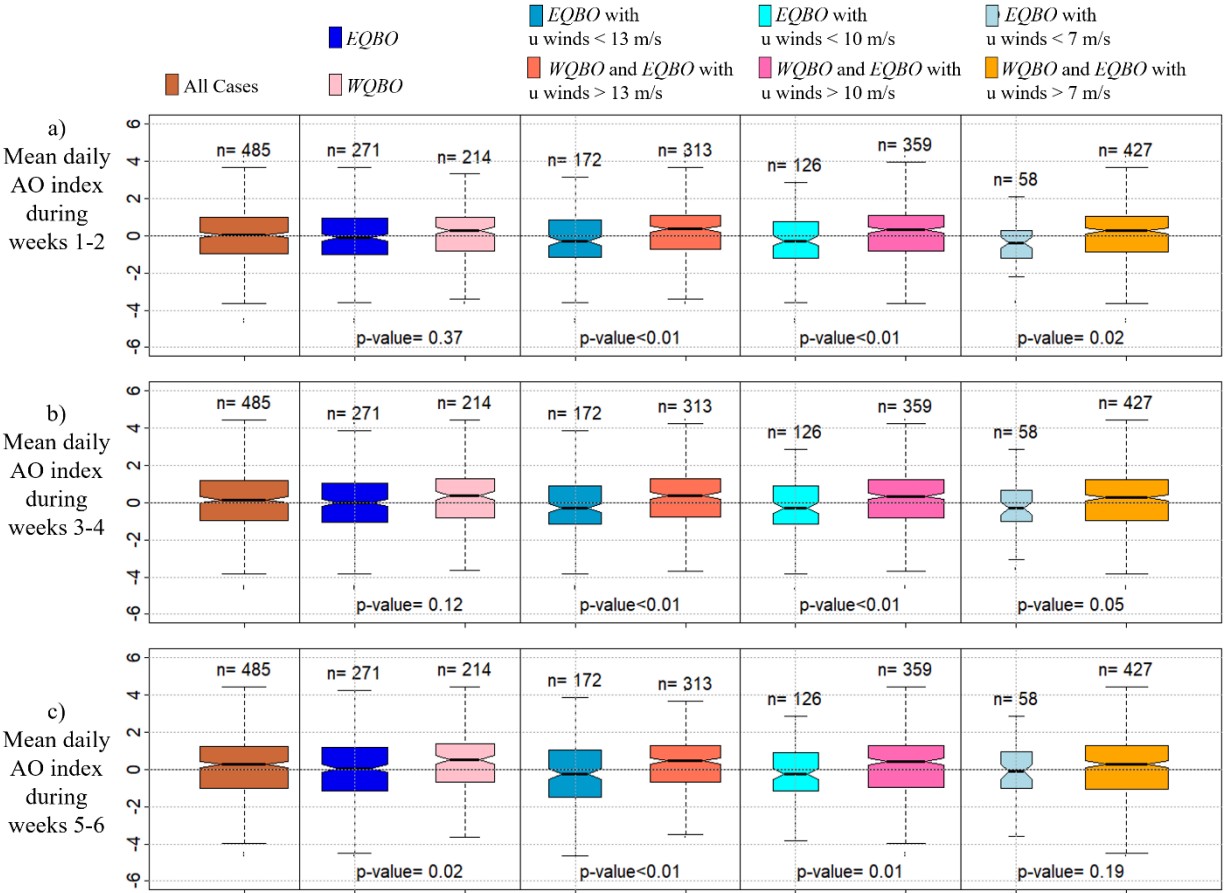

Figure 3: Observed daily AO index in November-March (1981-2016) a) 1–2, b) 3–4 and c) 5–6 weeks after different stratospheric situations. The horizontal line dividing each box into two parts shows the median of the data, the ends of the box show the lower and upper quartiles, and the whiskers represent the highest and the lowest values excluding outliers. The *n* written above each box indicates the number of observations in each group. The widths of the boxes have been drawn proportional to the square-roots of *n*. The p-value written below each boxplot pair indicates the likelihood of such a pair of distributions arising from a random sampling of a single distribution as given by a Student's t-test, i.e., p-values less than 0.05 indicate that the means of the data sets differ significantly at the 95% level of confidence. The notches of each side of the boxes were calculated by R boxplot.stats. If the notches of two plots do not overlap, this is 'strong evidence' that the two medians differ (Chambers et al., 1983, p. 62). ZMZW=zonal mean zonal wind at 60°N and 10 hPa. u winds=the QBO's monthly mean zonal wind components in levels 70 hPa…10 hPa.

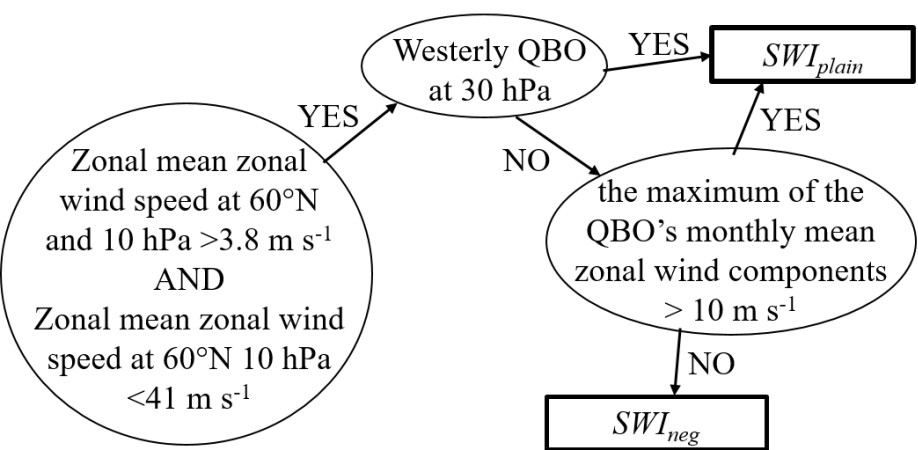

**Figure 4: Decision tree of *SWI*<sub>neg</sub>/*SWI*<sub>plain</sub>.**

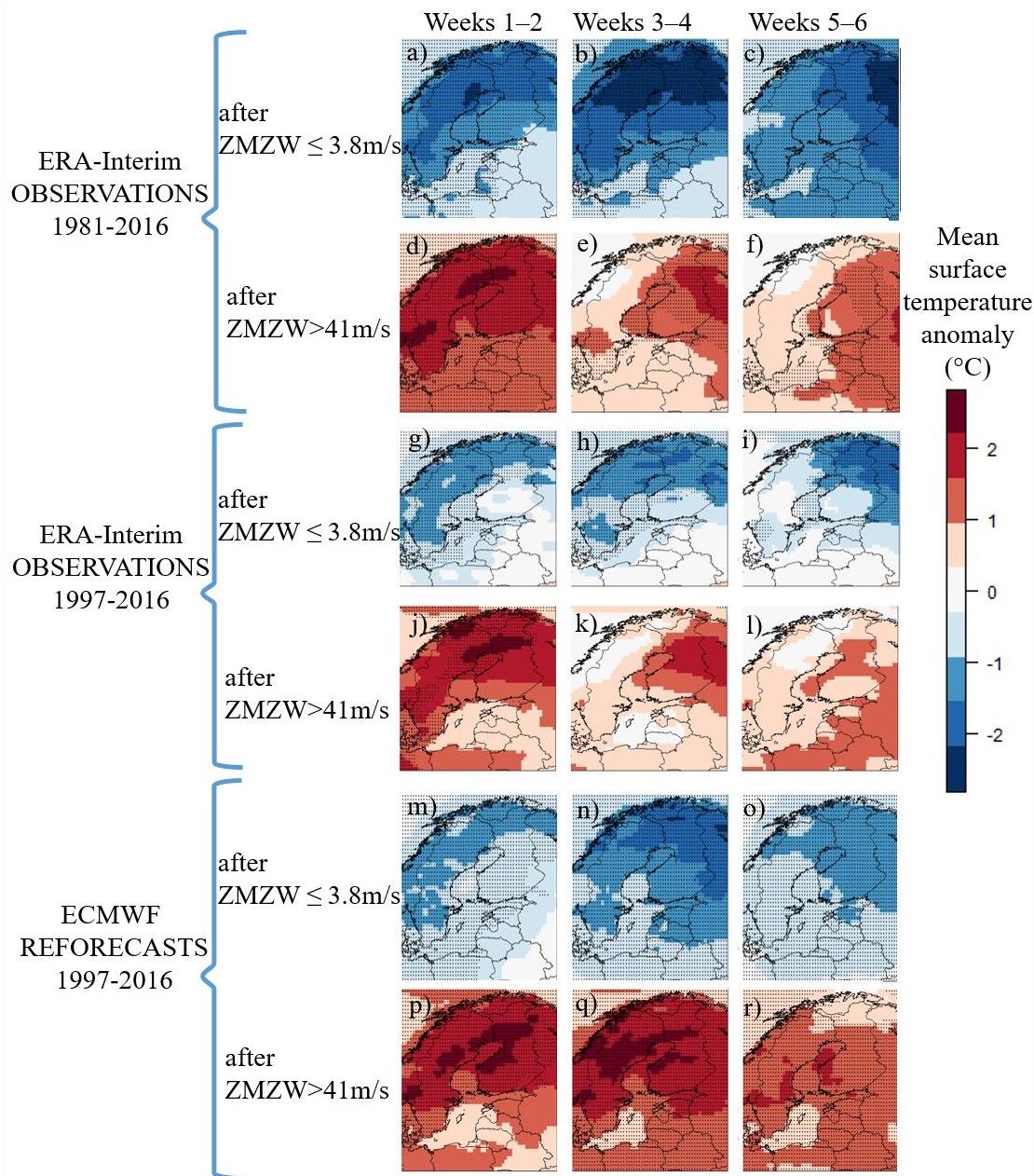

**Figure 5. ERA-Interim observed (a-l) and ECMWF reforecasted (m-r) mean temperature anomalies in comparison to the 1981-2016 mean during boreal winters (November-February) in cases the minimum of the daily mean zonal mean zonal wind (ZMZW) at 60°N and 10 hPa during the previous 10 days was below 3.8 ms-1 (covering about 11% of the winter weeks) or above 41 ms-1 (covering about 9% of the winter weeks). The dotted areas represent the 95% level of confidence where the means of surface temperature anomalies differ significantly from the rest of the cases.**

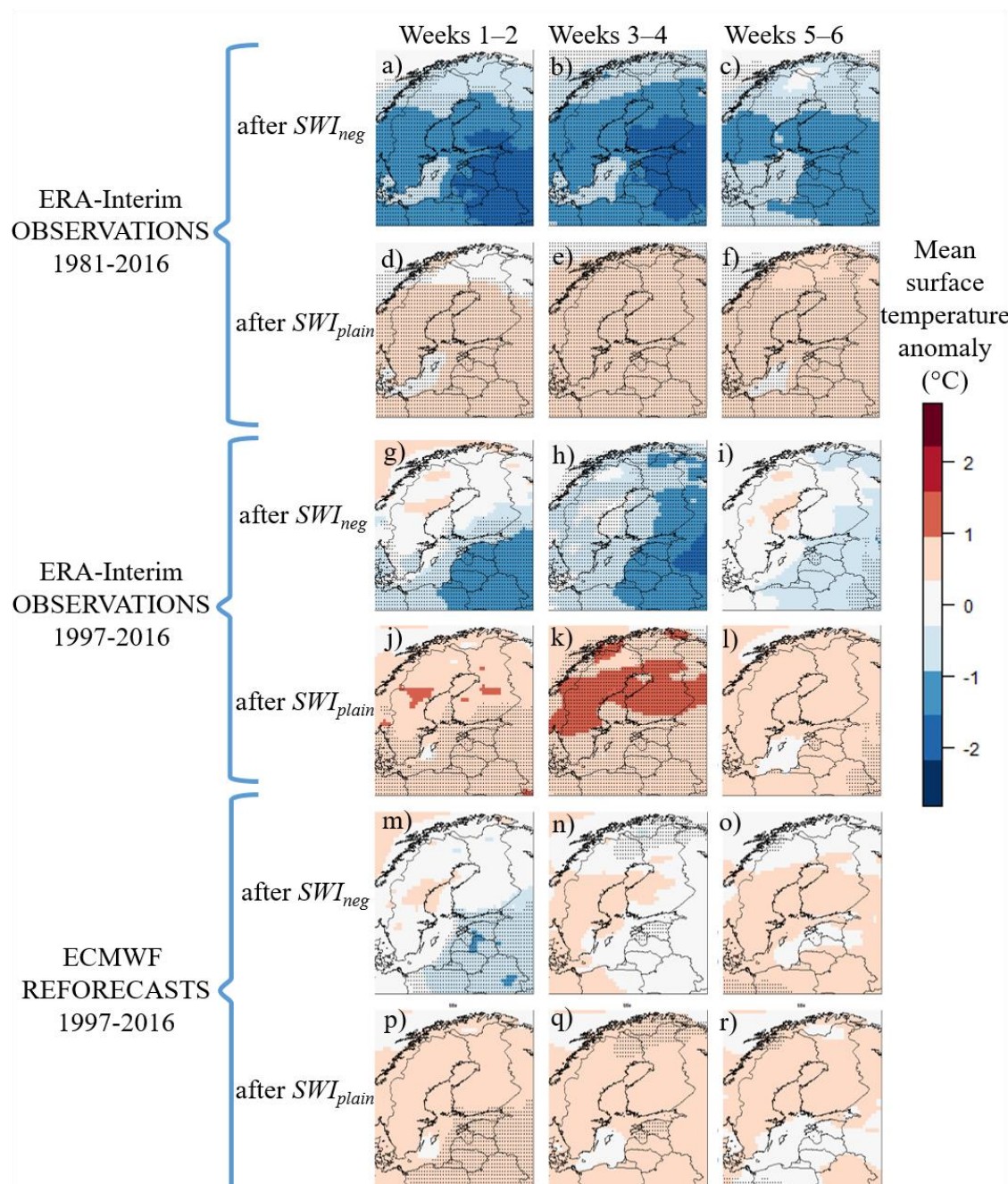

**Figure 6. ERA-Interim observed (a-l) and ECMWF reforecasted (m-r) mean temperature anomalies in comparison to the 1981-2016 mean during boreal winters (November-February) in cases the previous week's *SWI* was negative (*SWIneg*, covering about 21% of the winter weeks) or plain (*SWIplain*, covering about 59% of the winter weeks). The dotted areas represent the 95% level of confidence where the means of surface temperature anomalies after *SWIneg* and *SWIplain* differ significantly.**

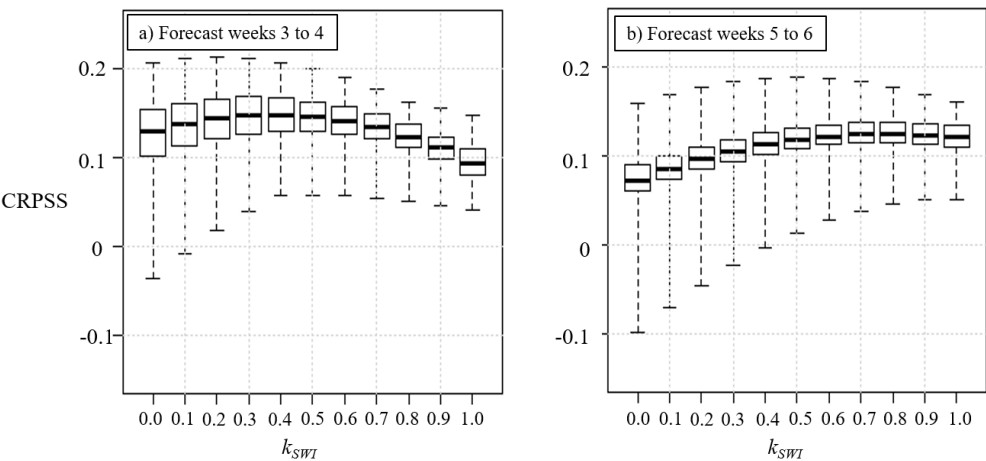

**Figure 7. Sensitivity of the expected CRPSS of the post-processed ECMWF surface temperature reforecasts to the $k_{SWI}$ ranging from 0.0 to 1.0 in forecast weeks 3–4 (a) and 5–6 (b). The black boxes show the lower and upper quartiles, and the whiskers illustrate the extremes of the November-February mean CRPSSs of all the grid points in Northern Europe.**

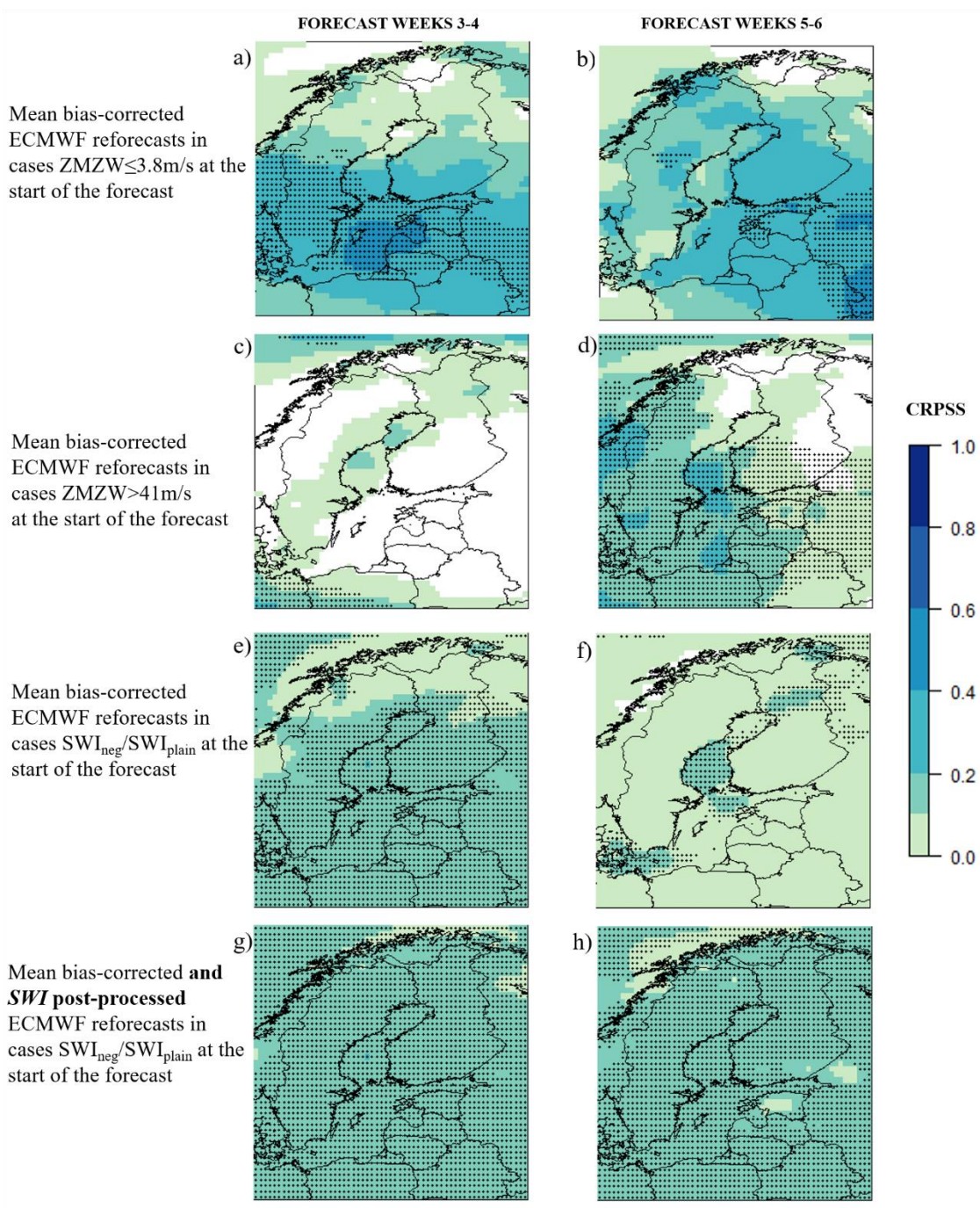

**Figure 8. Expected CRPSS of forecast weeks 3–4 and 5–6 of the ECMWF's mean temperature reforecasts for November–February 1997–2016 after mean bias-correction (a-f) and after both mean bias-correction and the *SWI* based post-processing (g-h). ERA-Interim climatology of 1981–2010 was used as the reference. The dotted areas represent the 95% level of confidence that the CRPSS is above zero.**