# Peer review of "Adding value to Extended-range Forecasts in Northern Europe by Statistical Post-processing Using Stratospheric Observations"

_Atmospheric Chemistry and Physics, 2019_

## Referee Comment (RC1) · Anonymous Referee #1 · 16 Sep 2019

The authors present a genuinely interesting analysis that contains new methods to improve extended-range forecasts. The great improvement in forecast skill must be useful work. I found the manuscript interesting and believe others would as well, but I have several questions and comments in the current form.

Major comments 1. The authors used 'minimum daily AO index' but it seems that there is no clear justification for the use of 'minimum'. Use of the minimum AO index might have more uncertainty because the value fluctuates with a day. The uncertainty would be reduced if the authors use weekly mean value rather than the 'minimum'. It would be helpful to isolate the significant skill increase from sampling issues. Additionally,

the post-processing revises weekly mean temperature. This is an additional reason to justify why we need 'minimum' AO index rather than weekly mean.

2. I am not sure the how the QBO can modulate AO index at weekly time scale. The QBO has an average period of ∼28 month. The QBO phase tends to prevail for the entire season so how can we connect dynamics between weekly variation of the AO and QBO?

3. The authors suggested that great improvement in forecast skill associated with QBO-polar vortex connection. The past studies suggested that the EQBO could modulate polar vortex, which in turn lead the AO. However, EQBO and polar vortex is not much coincided (Fig. 2). The number of EQBO (u wind <10m/s) is 34 and week vortex (ZMZW <3.8m/s) is 9. Sum of them is 43 but the number of SWIneg cases are 41, which means the events satisfying both condition is only 2. This implies that there should be relationship between EQBO and AO, which is independent to polar vortex. The authors should elaborate introduction and discussion for the prediction skill source for the statistical post-processing.

Minor comments The annotation "Fig. 2 EQBO and vortex ZMZW < 3.8m/s" (green) does not correspond to decision tree in Figure. 3. Please revise it for the better understanding.

---

## Referee Comment (RC2) · Anonymous Referee #2 · 16 Sep 2019

Summary

The authors use a post-processing technique based on stratospheric predictors of the Arctic Oscillation on ECMWF forecasts to try to improve predictive skill of surface temperature at weeks 3-6. Overall the paper addresses relevant scientific questions (given the lack of some stratospheric processes such as the QBO and its teleconnections in the forecast model), but certain aspects of the paper could be clarified and re-organized. I also have questions about their particular technique of only using forecasts made on the first week of each month, and using stratospheric data at 10 hPa, rather than in the lower stratosphere which is a better indicator of stratosphere-troposphere

coupling. I suggest a major revision.

General Comments

1) More could be explained up front about how the QBO teleconnection processes in particular are not well captured in the forecast models (particularly after early winter, as per Garfinkel et al. 2018), and why. In particular, see Line 14-15, page 3: Here it should be specifically mentioned whether the ECMWF S2S model (the version used here) is able to self-generate QBO variability (most S2S models cannot- instead they are initialized with observed QBO winds and degrade relatively quickly away from that towards model climatology). This should be further emphasized (possibly showing something like the forecasted QBO winds in the tropics compared to observed values for each of the initialization dates used here, so it's clear that the model is missing this process).

2) Sensitivity testing to the parameters chosen here could be further provided, as I'm not sure I understand what was the motivation for some of these choices. For example, see Line 21-25, page 5: Why were QBO winds at 30 hPa selected and was the sensitivity to this level tested? What about 40 or 50 hPa? Also, I understand using the 60N 10 hPa metric for stratospheric variability, but it's not a great metric for coupling to the surface. I assume you might see much better results using winds at a lower level in the stratosphere, near 50-100 hPa. See for example Karpechko et al. 2017 (https://doi.org/10.1002/qj.3017). What happens if you use a lower level instead?

3) Furthermore, presumably this technique of identifying weak vortex periods (Line 25, page 5) misses quite a few observed SSW events in the stratosphere (because they don't all occur in the last 10 days of the month). Could you say something about this, and why your method is still valuable? It also seemed like the sample size for u<3.8 m/s was extremely small (n=9) for the longer record period and must be even smaller for the shorter 1997-2016 period of the hindcasts. I wondered why it was so small I wondered why it was so small (144 forecasts, *10th percentile climatology = 14 times

this should happen). Can much be said for a predictor that only happens 9 times in 36 years? What happens if this threshold is relaxed to include more events? This is another reason why getting daily tropical winds so all forecast initialization dates may be used could be valuable. Along these lines, I'm confused about the number of events shown in Figure 2 for the two rightmost columns.

4) Finally, I'm curious about the choice of predictors in terms of their covariability. How many times when you had EQBO (and EQBO with maximum thresholds) did you also see a weak polar vortex at 60N 10mb? I assume these are correlated/concurrent, and the EQBO merely adds additional samples where the vortex is weak but not weak enough to meet the 3.8 m/s threshold. It would be interesting to state what the mean value of 60N 10mb zonal winds are during the various EQBO thresholds.

Specific Comments

1) Line 6, page 4: You don't consider forecasts in the summer here though, so why 52 weeks? Shouldn't it just be Nov-Feb forecasts? (also, because it wasn't clear- I assume you mean throughout that you use any runs initialized in Nov-Feb, but the forecasts in Feb obviously forecast into March/April, correct?)

2) Line 19-20, page 4: Are the operational forecasts used in this study? I'm not sure I understand why the CRPS is adjusted for 51 members if only the hindcasts with 11 members are shown throughout, but maybe I missed where the operational forecasts were used.

3) Line 23, page 4; line 2, page 5: Not sure what is meant here by "annual mean"- do you mean the average of weeks 1-6 across all years? Or the average across all months to get one value for each year? I'm not sure I follow what is meant in this paragraph. Also, you might present Figure 1 in section 2.1 and 2.2 in relation to what is being discussed to make it clearer.

4) Line 12-13, page 5: Isn't this data MERRA-2 reanalysis? Could you be more specific

about what this data is, and how it was derived? Also, if you are using ERA-interim to verify the forecasts- why not also use ERA-interim for daily zonal winds, both in the stratosphere polar vortex and in the tropics for the QBO? Singapore winds may not always represent the zonal-mean tropical winds that drive QBO teleconnections. Presumably the forecast model is initialized using the winds in the reanalysis, correct? And if you need daily winds to be able to look at forecasts other than the first forecast initialized each month, you could easily get this data all from one product, rather than three different products.

5) Line 18-30, page 5: I found this a bit hard to follow without any visual explanation, and I wonder if it would be clearer to discuss Figures 2 and 3 in this section instead of later.

6) Line 13, page 7: is CRPSS above zero a reasonable metric of skill? CRPSS near zero but positive surely can't be that useful (some of these values in Figure 1 are less than 0.1).

7) Line 19, page 7: Why was only the minimum daily AO considered and not the mean? Does the mean not change enough?

8) Line 25, page 8: I think Fig 4p looks very much like Fig 4j.

9) Figure 6: How are panels (a,b) different than Figure 1? (other than being two week averages rather than 1 week).

Technical Edits

1) Line 19, page 1; line 22-23, page 3; possibly other locations: specify that you are referring to the previous months' tropical stratospheric wind observations.

2) Line 2, page 2: not sure what is meant by "experimented during a one year living lab"

3) Line 3, page 2: put comma after "production"

4) Line 24, page 2: maybe instead "other definitions have been used"

5) Line 10, page 3: I would clarify that this paper looked at S2S hindcasts similar to what you are looking at here

6) Line 2, page 4: remove the word "scale"

7) Figure 1- dots seem a little blurry, might make sure it's high enough resolution for final version.

8) Line 29, page 7: add in "maximum" to "QBO's monthly mean zonal wind components" or it doesn't make much sense

9) Line 5, page 8: add in "where" between "cases" and "the"

10) Figure 2- might be nice to put in bold those p-values that are less than 0.05. Also, shouldn't the last column be labeled "EQBO with u winds < 10 m/s OR ZMZW at 60N and 10 hPa >3.8 m/s"?

11) Line 24, page 9: The way this is written is confusing, do you mean p decreased so significance increased?

12) Line 29, page 9: add a "to" after "corresponding"

---

## Referee Comment (RC3) · Anonymous Referee #3 · 19 Sep 2019

This paper describes a post-processing method to improve sub-seasonal forecasts of northern European winter temperatures, based on the state of the stratospheric polar vortex and QBO in the period immediately preceding the forecast. The paper is interesting, topical and clearly explained, but I have some reservations about the method that need to be addressed before the paper is suitable for publication.

Major comments

1. I'm not convinced about the inclusion of the QBO as a predictor of the Arctic Oscillation (AO) in the method. The authors note that the westerly / easterly QBO is associated with a stronger / weaker polar vortex, but the polar vortex is already included

as the other predictor. Unless the QBO directly influences the AO independently of the polar vortex, it's hard to see how the QBO can provide additional skill in forecasting the AO.

The authors state that the results are more significant when both the polar vortex and the QBO are used as predictors of the AO, presumably referring to the results in figure 2. The method partitions the 144 observed winter months (for NDJF, 1981-2016) into two sets based on a stratospheric precursor criterion (eg the polar vortex winds are either anomalously weak, or not). The aim is to make the two sets as distinct as possible in terms of their AO index values. Figure 2 shows the distribution of the AO values for each pair of sets obtained using various different criteria.

The partition based on the polar vortex and the QBO (green and purple boxes) does show marginally lower p-values than the partition based on the polar vortex alone (yellow and red boxes) consistent with the authors' claim. However, the partition based on the polar vortex alone is split 9 months to 135 months, leading to quite a large uncertainty in the mean AO-index value for the set of 9 months. I suspect this is leading to a higher p-value. Were other thresholds for the polar vortex winds tried, other than 3.8m/s?

In the figure, the difference between the median values appears larger for the partition based on the polar vortex alone (yellow and red boxes) than for the partition based on the polar vortex and QBO (green and purple boxes). This suggests to me that the QBO isn't obviously adding any skill in discriminating between high AO and low AO winters.

2. The method is based on partitioning the winter months into

i) those with an anomalously weak polar vortex and/or easterly QBO, and ii) all the remaining winter months

It's not really obvious how this method was arrived at. Have the authors considered also separating out the set of winter months with an anomalously strong polar vortex?

It seems like an obvious thing to try, and may provide additional skill in predicting the AO.

Minor comments

p2, line 26: How exactly is the Arctic Oscillation index defined? The authors just say it's based on 1000hPa geopotential height for 20-90N.

p4 line 19: "the CRPS_RF of the CRPS_rf" - what does this mean?

p6 lines 15-22: I didn't entirely follow the method for making anomalies here - if you're just taking the mean of the 7 anomalies based on different years, aren't you going to get the same answer as just using all the years?

p7 line 19: The method defines the AO value as the lowest value of the daily AO index in different weeks of the forecast. Why was this chosen - is it representative of the northern European temperature in those weeks? The weekly mean AO value would presumably be less noisy.

p8 line 6: the zonal mean zonal wind threshold is stated as 4.8m/s here, but 3.8m/s in figure 2.

---

## Author Comment (AC1) · 25 Nov 2019

**Authors Response to Interactive comments on "Adding value to Extended-range Forecasts in Northern Europe by Statistical Post-processing Using Stratospheric Observations" by Natalia Korhonen et al.**

The comments are in Black and the responses in blue.

We thank the reviewers for their thoughtful and constructive comments.

We have done several major changes to the manuscript. First, we have examined the mean AO instead of the minimum AO. In this process the Figure 2 was replotted. Second, we have increased the sample size of forecasts and observations by including all cases in Nov-Feb 1981-2016 (not just the cases after the first Monday in each month). In this process the Figures 2, 4, 5, and 6 were replotted. Third, we have in this author response demonstrated the ZMZW at 60 °N and 10 hPa at the start of the forecast and 1-6 weeks after different stratospheric situations. In addition to these, we have done several editings to the manuscript, to clarify it, according to the comments. Below we respond to the reviewers point-by-point.

Best regards

Natalia Korhonen and co-authors

**Anonymous Referee #1

The authors present a genuinely interesting analysis that contains new methods to improve extended-range forecasts. The great improvement in forecast skill must be useful work. I found the manuscript interesting and believe others would as well, but I have several questions and comments in the current form.

Major comments

1. The authors used 'minimum daily AO index' but it seems that there is no clear justification for the use of 'minimum'. Use of the minimum AO index might have more uncertainty because the value fluctuates with a day. The uncertainty would be reduced if the authors use weekly mean value rather than the 'minimum'. It would be helpful to isolate the significant skill increase from sampling issues. Additionally, the post-processing revises weekly mean temperature. This is an additional reason to justify why we need 'minimum' AO index rather than weekly mean.

As suggested, we have now used the "mean" instead of "minimum" in the current manuscript.

2. I am not sure the how the QBO can modulate AO index at weekly time scale. The QBO has an average period of ~28 month. The QBO phase tends to prevail for the entire season so how can we connect dynamics between weekly variation of the AO and QBO?

QBO gives only monthly impact on the probability forecast, however, the ZMZW at 60 °N at 10 hPa (indicating the strength of the polar vortex) gives weekly impact as it is using the last 10 days preceding the start of the forecast.

3. The authors suggested that great improvement in forecast skill associated with QBO polar vortex connection. The past studies suggested that the EQBO could modulate polar vortex, which in turn lead the AO. However, EQBO and polar vortex is not much coincided (Fig. 2). The number of EQBO (u wind <10m/s) is 34 and week vortex (ZMZW <3.8m/s) is 9. Sum of them is 43 but the number of SWIneg cases are 41, which means the events satisfying both condition is only 2. This implies that there should be relationship between EQBO and AO, which is independent to polar vortex. The authors should elaborate introduction and discussion for the prediction skill source for the statistical post-processing.

In the current revised manuscript we increased the sample size by including all cases in Nov-Feb 1981-2016 (not just the cases after the first Monday in each month). Thereby the number of cases, n, in Figure 2 is now higher than in the discussion paper. We plotted Figure AR1 (in this document) to demonstrate how the mean ZMZW at 60 °N 10 hPa was at the start of *EQBO*, *WQBO* etc. (Fig. AR1 a) and 1-6 weeks after *EQBO*, *WQBO* etc. (Fig. AR1 b-d). Fig. AR1 a) shows that the weaker than 3.8m/s ZMZW does indeed not often coincide with the *EQBO* etc. at the start of the forecast. However, Fig. AR1 b-d shows that the mean ZMZW at 60 °N 10 hPa is lower 3-6 weeks after *EQBO* than after *WQBO*.

Editings to Introduction:

"Scaife et al. (2014, Fig.4a) demonstrated indicators of a more negative AO in the easterly QBO at level 30 hPa than in the westerly QBO phase at this level."

Editings to Discussion:

"We investigated the prediction source of the QBO at 30 hPa. In line with Scaife et al. (2014) we found that AO was weaker 1-6 weeks after EQBO that WQBO at 30 hPa. As negative AO index enables cold air outbreaks to Northern Europe (Thompson et al. 2002, Tomassini et al. 2012) and positive AO index tends to bring milder and wetter than average weather to Northern Europe (Limpasuvan et al. 2005), we tested the predictor $SWI_{neg}$/$SWI_{plain}$ as a predictor of mean surface temperature in Northern Europe for forecast weeks 3-6. We found that the mean surface temperature anomalies in Northern Europe in November–February in 1981–2016 after SWIneg and SWIplain were statistically significantly different, with anomalously cold surface temperatures more common 3–6 weeks after SWIneg."

**Mean ZMZW (ms⁻¹) at 60°N and 10 hPa**

[Figure]

*Figure AR1. Mean Zonal Mean Zonal Wind (ZMZW) at 60N and 10 hPa. The red, black and green vertical lines represent the third quantile, the median, and the first quantile, respectively, of the ZMZW at 60 °N and 10 hPa in November-March 1981-2016. The horizontal line dividing each box into two parts shows the median of each data, the ends of the box show the lower and upper quartiles, and the whiskers represent the highest and the lowest values excluding outliers. The n written above each box indicates the number of observations in each group. The widths of the boxes have been drawn proportional to the square-roots of n. The notches of each side of the boxes were calculated by R boxplot.stats. If the notches of two plots do not overlap, this is 'strong evidence' that the two medians differ (Chambers et al., 1983, p. 62). ZMZW=zonal mean zonal wind. SWI=Stratospheric Wind Index.*

Minor comments

The annotation "Fig. 2 EQBO and vortex ZMZW < 3.8m/s" (green) does not correspond to decision tree in Figure. 3. Please revise it for the better understanding.

We removed the colors from Figure 3 to avoid misunderstanding.

**Anonymous Referee #2

Summary

The authors use a post-processing technique based on stratospheric predictors of the Arctic Oscillation on ECMWF forecasts to try to improve predictive skill of surface temperature at weeks 3-6. Overall the paper addresses relevant scientific questions (given the lack of some stratospheric processes such as the QBO and its teleconnections in the forecast model), but certain aspects of the paper could be clarified and reorganized. I also have questions about their particular technique of only using forecasts made on the first week of each month, and using stratospheric data at 10 hPa, rather than in the lower stratosphere which is a better indicator of stratosphere-troposphere coupling. I suggest a major revision.

General Comments

1) More could be explained up front about how the QBO teleconnection processes in particular are not well captured in the forecast models (particularly after early winter, as per Garfinkel et al. 2018), and why. In particular, see Line 14-15, page 3: Here it should be specifically mentioned whether the ECMWF S2S model (the version used here) is able to self-generate QBO variability (most S2S models cannot- instead they are initialized with observed QBO winds and degrade relatively quickly away from that towards model climatology). This should be further emphasized (possibly showing something like the forecasted QBO winds in the tropics compared to observed values for each of the initialization dates used here, so it's clear that the model is missing this process).

The version of the ECMWF used here is IFS cycle 43r1. The representation of the QBO in this version of the IFS is described in detail in Johnson et al. (2019) and Stockdale et al (2018). The skill of the QBO forecasts decreases substantially after the first 2 months of the forecast. This is shown to be sensitive to the parametrization of the tropical non-orographic gravity wave drag in the model (see also Polichtchouk et al. (2018), Polichtchouk et al. (2017)). The amplitude of the QBO tends to weaken through the forecast. All the forecasts in our study are initialised from ERA-Interim reanalysis: the QBO is well represented in the initial conditions but as noted the amplitude tend to weakens during the 6 weeks of the forecast. This is shown in Garfinkel et al (2018) and we have revised the text to clarify this.

Johnson, S. J., Stockdale, T. N., Ferranti, L., Balmaseda, M. A., Molteni, F., Magnusson, L., Tietsche, S., Decremer, D., Weisheimer, A., Balsamo, G., Keeley, S. P. E., Mogensen, K., Zuo, H., and Monge-Sanz, B. M.: SEAS5: the new ECMWF seasonal forecast system, Geosci. Model Dev., 12, 1087–1117, https://doi.org/10.5194/gmd-12-1087-2019, 2019.

Polichtchouk, I., et al., 2017: What influences the middle atmosphere circulation in the IFS? ECMWF Technical Memorandum No. 809.

Stockdale, T. et al., 2018. SEAS5 and the future evolution of the long-range forecast system, ECMWF Technical Memoranda n. 835. DOI: 10.21957/z3e92di7y

2) Sensitivity testing to the parameters chosen here could be further provided, as I'm not sure I understand what was the motivation for some of these choices. For example, see Line 21-25, page 5: Why were QBO winds at 30 hPa selected and was the sensitivity to this level tested? What about 40 or 50 hPa? Also, I understand using the 60N 10 hPa metric for stratospheric variability, but it's not a great metric for coupling to the surface. I assume you might see much better results using winds at a lower level in the stratosphere, near 50-100 hPa. See for example Karpechko et al. 2017 (https://doi.org/10.1002/qj.3017). What happens if you use a lower level instead?

QBO winds at 30 hPa were chosen because we found better effect on the mean AO in the coming weeks then by the 40 hPa or 50hPa. Also Scaife et al. (2014) demonstrated in their Fig. 4a that the AO was more negative after easterly QBO at 30 hPa that westerly QBO at the same level. As we are using the mean QBO winds of the previous month as predictors, this means that in the 5-6 weeks forecast the QBO observation is 2 to 2.5 months old. Indeed the 30 hPa values in time precede the 40 hPa and 50 hPa.

Karpechko et al. 2017 found that the conditional probability of having a tropospheric signal after SSW depends on the value of AO at 150 hPa during 0-4 days after the ZMZW at 60 N and 10 hPa had turned easterly (central date=CD). We are not able to directly implement this to our post-processing, as we are not using the CD, but for our post-processing of the probabilistic forecasts we are using the ZMZW at 60 N and 10 hPa being below its November-March 10th percentile (3.8m/s) as an indicator for a weak polar vortex at the start of the forecast and probable a more negative surface AO index in the coming 1-6 weeks.

3) Furthermore, presumably this technique of identifying weak vortex periods (Line 25, page 5) misses quite a few observed SSW events in the stratosphere (because they don't all occur in the last 10 days of the month). Could you say something about this, and why your method is still valuable? It also seemed like the sample size for u<3.8 m/s was extremely small (n=9) for the longer record period and must be even smaller for the shorter 1997-2016 period of the hindcasts. I wondered why it was so small I wondered why it was so small (144 forecasts, *10th percentile climatology = 14 times this should happen). Can much be said for a predictor that only happens 9 times in 36 years? What happens if this threshold is relaxed to include more events? This is another reason why getting daily tropical winds so all forecast initialization dates may be used could be valuable. Along these lines, I'm confused about the number of events shown in Figure 2 for the two rightmost columns.

In the revised manuscript we have increased the sample size by including all cases in Nov-Feb 1981-2016 (not just the cases after the first Monday in each month). Now the sample size for u<3.8 m/s is 70 for the 1981-2016 period.

4) Finally, I'm curious about the choice of predictors in terms of their covariability. How many times when you had EQBO (and EQBO with maximum thresholds) did you also see a weak polar vortex at 60N 10mb? I assume these are correlated/concurrent, and the EQBO merely adds additional samples where the vortex is weak but not weak enough to meet the 3.8 m/s threshold. It would be interesting to state what the mean value of 60N 10mb zonal winds are during the various EQBO thresholds.

Figure AR1 above in this document shows boxplots of the mean ZMZW at 60 °N 10 hPa at the start of *EQBO*, *WQBO* etc. (Fig. AR1 a) and 1-6 weeks after *EQBO*, *WQBO* etc. (Fig. AR1 b-d). Figure AR1 shows that the mean ZMZW at 60 °N 10 hPa is weaker 3-6 weeks after *EQBO* and *EBOQ* with different thresholds (Fig. AR1 c-d) than at the start of the forecast (Fig. AR1 a).

Specific Comments

1) Line 6, page 4: You don't consider forecasts in the summer here though, so why 52 weeks? Shouldn't it just be Nov-Feb forecasts? (also, because it wasn't clear I assume you mean throughout that you use any runs initialized in Nov-Feb, but the forecasts in Feb obviously forecast into March/April, correct?)

In this part and in Figure 1 we have actually verified reforecasts run for every week (52 weeks) of years 1997-2016 (20 years), i.e., 52*20=reforecasts 1040. The *SWI* post-processing, however, was done only for reforecasts initialized in Nov-Feb. And correct, the forecast initialized in February forecast into March/April and those are also included in the post-processing.

2) Line 19-20, page 4: Are the operational forecasts used in this study? I'm not sure I understand why the CRPS is adjusted for 51 members if only the hindcasts with 11 members are shown throughout, but maybe I missed where the operational forecasts were used.

There are no operational forecasts used in this study. We refer to the operational forecasts of the ECMWF's IFS which reforecasts we verify. We added "of the ECMWF's IFS" after "the operational forecasts".

3) Line 23, page 4; line 2, page 5: Not sure what is meant here by "annual mean"- do you mean the average of weeks1-6 across all years? Or the average across all months to get one value for each year? I'm not sure I follow what is meant in this paragraph. Also, you might present Figure 1 in section 2.1 and 2.2 in relation to what is being discussed to make it clearer.

We edited this part by:

We calculated the annual means of the expected $CRPS_{RF}$ across all weeks (1 to 52) of the years 1997-2016 reforecasts. These annual means were computed separately for lead times of 1 week, 2 weeks, 3 weeks, 4 weeks, 5 weeks, and 6 weeks, here called forecast week 1, forecast week2, forecast week 3, forecast week 4, forecast week 5, and forecast week 6, respectively. Further, the skill scores of the annual mean CPRSs, the annual mean CRPSSs, for each lead time were calculated as follows:

$$CRPSS = 1 - \frac{CRPS_{RF}}{CRPS_{clim}} \qquad (3).$$

4) Line12-13, page 5: Isn't this data MERRA-2 reanalysis? Could you be more specific about what this data is, and how it was derived? Also, if you are using ERA-interim to verify the forecasts-why not also use ERA-interim for daily zonal winds, both in the stratosphere polar vortex and in

the tropics for the QBO? Singapore winds may not always represent the zonal-mean tropical winds that drive QBO teleconnections. Presumably the forecast model is initialized using the winds in the reanalysis, correct? And if you need daily winds to be able to look at forecasts other than the first forecast initialized each month, you could easily get this data all from one product, rather than three different products.

Yes, this is MERRA-2 reanalysis, we added this information to the manuscript.

In the current manuscript we increased the sample size by including all cases in Nov-Feb 1981-2016 (not just the cases after the first Monday in each month). For QBO we still used the Singapore winds, we just used the previous months' Singapore winds for every weeks' forecast. For the ZMZW at 60N and 10hPa we always used the most recent, the last 10 days' wind reanalysis data. Hence this lead to variation in *SWI* even within the month with constant QBO.

5) Line 18-30, page 5: I found this a bit hard to follow without any visual explanation, and I wonder if it would be clearer to discuss Figures 2 and 3 in this section instead of later.

To clarify what was done, we edited this part by:

As Scaife et al. (2014) demonstrated a more negative AO in the easterly QBO at 30 hPa compared to the westerly QBO at 30 hPa, we explored the AO index 1–6 weeks after following predictors:

- westerly QBO at 30 hPa, the *WQBO*,
- easterly QBO at 30 hPa, the *EQBO*,
- *EQBO* with the maximum of the monthly mean zonal wind components of the QBO between 70 hPa and 10hPa restricted to $7ms^{-1}$, $10ms^{-1}$, and $13ms^{-1}$,
- the daily ZMZW at 60° N and 10 hPa during the last 10 days of the previous month falling below its overall wintertime (November–March 1981–2016) $10^{th}$ percentile, corresponding a value of 3.8m/s, indicating a weak polar vortex already at the start of the forecast.

The statistical significance of the difference between the AO index following two different stratospheric situations, e.g., the *EQBO* and the *WQBO*, was determined using a two-sided Student's t-test with the null hypothesis that there is no difference. The most statistically significant predictors for weaker AO indexes observed 1–2 weeks, 3–4 weeks, and 5–6 weeks after these stratospheric situations, were used to define a *SWI* to be *SWI_{neg}*; otherwise, it was defined as *SWI_{plain}* for the beginning of each winter month (November–February) in 1981–2016.

6) Line 13, page 7: is CRPSS above zero a reasonable metric of skill? CRPSS near zero but positive surely can't be that useful (some of these values in Figure 1 are less than 0.1).

CRPSS above zero means that this probability forecast is at least better that the climatological forecast (here the climatological forecast is a 30 member ensemble of 1981-2011 weekly mean surface temperature observations).

7) Line19, page7: Why was only the minimum daily AO considered and not the mean? Does the mean not change enough?

As suggested, we have now used the "mean" instead of "minimum" in the current manuscript.

8) Line 25, page 8: I think Fig 4p looks very much like Fig 4j.

Yes, and we edited the text to bring this up.

9) Figure 6: How are panels (a,b) different than Figure 1? (other than being two week averages rather than 1 week).

In Figure 1 the CRPSSs are means of the reforecasts of all weeks of years 1997-2016 (52 weeks, 20 years), whereas in Figure (a,b) only reforecast initialized for November-February 1997-2016 are included.

Technical Edits

1) Line 19, page 1; line 22-23, page 3; possibly other locations: specify that you are referring to the previous months' tropical stratospheric wind observations.

We edited the "observations" to be "conditions" as this includes (in addition to tropical stratospheric wind observations) also the MERRA-2 reanalysis of the zonal mean zonal wind (ZMZW) at 60 °N and 10 hPa.

2) Line 2, page 2: not sure what is meant by "experimented during a one year living lab"

"during a one year living lab" was edited to "by a one year piloting phase"

3) Line 3, page 2: put comma after "production"

Done.

4) Line 24, page 2: maybe instead "other definitions have been used"

Done.

5) Line 10, page 3: I would clarify that this paper looked at S2S hindcasts similar to what you are looking at here

We edited to be:

"Even though some S2S models, including the ECMWF's Integrated Forecasting System (IFS, Vitart, 2014), are already able to reproduce the QBO's effect on the polar vortex, they are still underestimating the effect on surface weather (Garfinkel et al. 2018)."

6) Line 2, page 4: remove the word "scale"

Done.

7) Figure 1- dots seem a little blurry, might make sure it's high enough resolution for final version.

Done.

8) Line 29, page 7: add in "maximum" to "QBO's monthly mean zonal wind components" or it doesn't make much sense

Done.

9) Line 5, page 8: add in "where" between "cases" and "the"

Done.

10) Figure 2- might be nice to put in bold those p-values that are less than 0.05. Also, shouldn't the last column be labeled "EQBO with u winds < 10 m/s OR ZMZW at 60N and 10 hPa >3.8 m/s"?

We labeled the last column as suggested.

11) Line 24, page 9: The way this is written is confusing, do you mean p decreased so significance increased?

Yes, here the "decrease" should have been "increase". This sentence was, however, removed while editing the text.

12) Line 29, page 9: add a "to" after "corresponding"

Done.

**Anonymous Referee #3

This paper describes a post-processing method to improve sub-seasonal forecasts of northern European winter temperatures, based on the state of the stratospheric polar vortex and QBO in the period immediately preceding the forecast. The paper is interesting, topical and clearly explained, but I have some reservations about the method that need to be addressed before the paper is suitable for publication.

Major comments

1. I'm not convinced about the inclusion of the QBO as a predictor of the Arctic Oscillation (AO) in the method. The authors note that the westerly / easterly QBO is associated with a stronger / weaker polar vortex, but the polar vortex is already included as the other predictor. Unless the QBO directly influences the AO independently of the polar vortex, it's hard to see how the QBO can provide additional skill in forecasting the AO. The authors state that the results are more significant when both the polar vortex and the QBO are used as predictors of the AO, presumably referring to the results in figure 2. The method partitions the 144 observed winter months (for NDJF, 1981-2016) into two sets based on a stratospheric precursor criterion (eg the polar vortex winds are either anomalously weak, or not). The aim is to make the two sets as distinct as possible in terms of their AO index values. Figure 2 shows the distribution of the AO values for each pair of sets obtained using various different criteria. The partition based on the polar vortex and the QBO (green and purple boxes) does show marginally lower p-values than the partition based on the polar vortex alone (yellow and red boxes) consistent with the authors' claim. However, the partition based on the polar vortex alone is split 9 months to 135 months, leading to quite a large uncertainty in the mean AO-index value for the set of 9 months. I suspect this is leading to a higher p-value. Were other thresholds for the polar vortex winds tried, other than 3.8m/s? In the figure, the difference between the median values appears larger for the partition based on the polar vortex alone (yellow and red boxes) than for the partition based on the polar vortex and QBO (green and purple boxes). This suggests to me that the QBO isn't obviously adding any skill in discriminating between high AO and low AO winters.

In the current manuscript we have increased the sample size by including all cases in Nov-Feb 1981-2016 (not just the cases after the first Monday in each month). Now the sample size for the partition based on the polar vortex alone is 70 (out of 612) for the 1981-2016 period giving more certainty in the mean AO index values. The difference between the median values are larger for the partition based on the polar vortex alone (yellow and red boxes) than for the partition based on the polar vortex and QBO (green and purple boxes), however the sample size for $SWI_{neg}$ (199) is larger than weak polar vortex alone (70) giving more certainty for using this in post-processing the probabilistic 3-4 and 5-6 weeks mean temperature forecasts.

2. The method is based on partitioning the winter months into i) those with an anomalously weak polar vortex and/or easterly QBO, and ii) all the remaining winter months It's not really obvious how this method was arrived at. Have the authors considered also separating out the set of winter months with an anomalously strong polar vortex? It seems like an obvious thing to try, and may provide additional skill in predicting the AO.

Yes, we tried this also, but so far we were not able to find predictors for the stronger AO that would have also improved the forecasting skills.

Minor comments

p2, line 26: How exactly is the Arctic Oscillation index defined? The authors just say it's based on 1000hPa geopotential height for 20-90N.

To clarify the AO, we added:

"In Northern Europe one of the important indicators of the large-scale weather patterns is the phase of the AO. The AO is a climate pattern characterized by winds circulating counter clockwise around the Arctic at around 55°N latitude."

p4 line 19: "the CRPS_RF of the CRPS_rf" - what does this mean?

the CRPS_RF is the expected CRPS (assuming there were 51 members) of the ECMWF's reforecast, and the CRPS_rf is the CRPS of the ECMWF's reforecast with 11 members. We edited the text to clarify this.

p6 lines 15-22: I didn't entirely follow the method for making anomalies here - if you're just taking the mean of the 7 anomalies based on different years, aren't you going to get the same answer as just using all the years?

Yes, we changed this and the text to use just the mean of all years.

p7 line 19: The method defines the AO value as the lowest value of the daily AO index in different weeks of the forecast. Why was this chosen - is it representative of the northern European temperature in those weeks? The weekly mean AO value would presumably be less noisy.

As suggested, we have now used the "mean AO" instead of "minimum AO" in the current manuscript.

p8 line 6: the zonal mean zonal wind threshold is stated as 4.8m/s here, but 3.8m/s in figure 2.

The threshold was here corrected to 3.8m/s.

---

## Author Response (AR2)

**Authors Response to comments on "Adding value to Extended-range Forecasts in Northern Europe by Statistical Post-processing Using Stratospheric Observations" by Natalia Korhonen et al.**

The comments are in Black and the responses in blue.

We thank the reviewers for their comments on the revised manuscript.

We have now done further changes to the manuscript.

First, we have shown the AO index after different strengths of the zonal mean zonal wind speed at 60°N and 10hPa (ZMZW). For this we plotted a new Figure 2.

Second, we added a brief description of how the AO index from the NCEP CPC is produced.

Third, we have separated both the strong and weak ZMZW cases from the post-processing to show that the post-processing adds skill to the forecast even without them. In course of this, we updated Figures 3 and 4, added a new Figure 5 and updated Figures 6, 7, and 8.

In addition to these, we have done further editing to the manuscript, to clarify it, according to the comments. Below we respond to the reviewers point-by-point.

Best regards

Natalia Korhonen and co-authors

**Anonymous Referee #2**
**Suggestions for revision**
Overall the authors have addressed my comments and I think the analysis has been strengthened by incorporating all the weeks in the analysis rather than just the first week of the month. I do have a few remaining minor concerns outlined below.

Minor comments
-I still don't find it that convincing that the QBO information adds anything. Could you compare in Figure 6, for example, what the skill is if SWIneg is only based on the ZMZW factor?

Yes, we have now separated the weak and strong ZMZW cases (at the start of the forecast) and shown that actually the CRPSS in the former Figure 6, now Figure 8, of weeks 5 to 6 is lower when cases of strong and weak ZMZW are left out (Fig. 8e-f). And the post-processing indeed shows to improve the CRPSS in the forecast where the ZMZW is non-weak and non-strong at the start of the forecast (Fig. 8g-h).

-Page 2, Line 27-I do not agree with the new description added of the AO as a "climate pattern characterized by winds circulating counterclockwise around the Arctic at around 55N latitude". The AO

characterizes fluctuations in atmospheric mass between the pole and mid-latitudes, and explains latitudinal shifts in the eddy-driven jet near 40N (see Thompson and Wallace 2000). The AO is not another word for "jet stream" or "polar front jet". The authors should also include a brief description of how the AO from the NCEP CPC is calculated in the methodology section.

In the Introduction (p.2) we edited the mentioned lines about the AO to:
"During boreal winter the strength of the polar vortex affects the phase of the AO, which characterizes air mass flow between the Arctic and the mid-latitudes."

In the methodology section we added the description of the AO to the manuscript:
"The daily AO index from the NCEP CPC is produced by projecting the daily 1000 hPa geopotential height anomalies north of 20 °N onto the loading pattern of AO, which is defined as the first leading mode from the Empirical Orthogonal Function (EOF) analysis of monthly mean 1000 hPa height anomalies poleward of 20 °N during 1979-2000."

-Page 1, line 28: "lead time up to 46 days"- why 46 days instead of 42 days, which would be 6 weeks? Is this a general definition for extended-range?

The ECMWF ERFs are run some days longer than six weeks, however less than seven weeks. As we are using weekly means, we cannot use the last extra-days.

-Page 3, line 9: cite Garfinkel et al. 2012

Cited Garfinkel et al. 2012.
Garfinkel, C. I., Shaw, T. A., Hartmann, D. L., & Waugh, D. W. (2012). Does the Holton-Tan mechanism explain how the quasi-biennial oscillation modulates the arctic polar vortex? Journal of the Atmospheric Sciences, 69(5), 1713–1733. https://doi.org/10.1175/JAS-D-11-0209.1

-Page 3, line 15: Is this comment on the skill of the QBO forecasts true just for ECMWF or for all ERF systems? It's unclear.

Mainly this paragraph refers to the ECMWF model. However, Garfinkel et al. 2018 showed that most S2S models have difficulty to predict the QBO weeks ahead. We edited the text to clarify this.

-Page 5, Line 9: why 5 weeks? Doesn't the analysis go to week 6?

The mean bias-correction is done separately for each of the six forecast weeks. For defining the mean bias, we used the mean bias of the 5 weeks centered on the forecast weeks date, i.e., the forecast week and ±2 weeks.

-Throughout: AO "indexes" should either just be "index" or "indices"

Ok. We edited them to be "index".

-Page 6, Line 3: here I think "month" should be "week", according to the new analysis of using all weeks, correct? Similarly, page 6, line 9-10 again refers to the "beginning of each winter month", as does page 6, lines 14-15 and page 6, line 22. Also see caption for Figure 4.

Yes, thank you, we edited these "months" to "weeks".

**Anonymous Referee #3**

**Suggestions for revision**

I have now read the revised paper and response to reviewers. Although most of my comments have been satisfactorily dealt with, I don't feel that my main reservation about the analysis has been adequately addressed.

Major comment

Given that the QBO is commonly thought to influence AO via the stratospheric polar vortex, I remain sceptical that the QBO provides additional skill in forecasting the AO index over the skill obtained from using the polar vortex strength alone (which the authors still don't seem to have tried).

I therefore suggest the authors try using the partition based on polar vortex strength (ZMZW index) only to forecast the AO. Unless they can demonstrate a significant improvement in forecast skill from also including the QBO in their SWI forecast model, the QBO should not be used. Why include an additional predictor if it isn't needed?

We have now in Figure 2 demonstrated the AO after different strengths of ZMZW at 60°N and 10 hPa. We have also partitioned the strong and weak ZMZW (at the start of the forecast) from the $SWI_{neg}$ and $SWI_{plain}$ and updated the mean temperature anomalies figures to have this separation as well. We found that yes the strong and weak ZMZW (at the start of the forecast) have actually potential to lead to better predictability than in cases of intermediate ZMZW. We also found (see Figure 8, compare Figures 8e and 8g and Figures 8f and 8h) that the post-processing improved the mean CRPSS also when both weak and strong ZMZW cases were left out.

The authors agree that in figure 2, the partitions based on the ZMZW index (yellow and red boxes) have a greater difference in the median values than the authors' preferred SWI index (green and purple boxes). The p-values in both cases are now less than 0.01.

I don't really follow the authors' comment that the larger partition size for their SWI_neg index (199 members, green box in figure 2) compared to the partition based on low ZMZW (70 members, yellow box in figure 2) gives 'more certainty' in post-processing. Surely the real test of whether to include the QBO in the AO forecast model is whether it improves the forecast skill score?

Now we did not include the weak ZMZW cases anymore in the post-processing and Figure 8 shows that the CRPSS was still improved.

Even if the relative lack of members in the low ZMZW partition is a problem (which isn't immediately obvious to me) more lenient threshold values of ZMZW could be tried to rectify this (as I suggested previously).

In Figure 2 we have now shown the AO after different thresholds of ZMZW. Also in temperature anomalies we have now examined the effect on the mean temperatures after strong and weak ZMZW (Fig.5). Moreover: not to get improvement for the post-processing from the strong and weak ZMZW we have now left those cases out of $SWI_{neg}$ and $SWI_{plain}$.

Minor comments

Regarding my previous major comment #2, I think it would be worth noting in the paper that looking at winters with a strong polar vortex has been tried, even though it didn't improve the AO forecast skill.

New figures Fig. 2 and Fig. 5d)-5f) and 5j)-5l) demonstrate that high ZMZW is on average followed by higher AO and surface temperature.
Further new figures Fig. 8a) and 8b) demonstrate the mean forecast skill (CRPSS) in cases the ZMZW is weaker than 3.8m/s and shows that for weeks 3-6 the mean CRPSS reached values even above 0.4.
Figure 8c) and 8d) demonstrate how the CRPSS is in cases ZMZW is stronger than 41m/s and shows that for weeks 5–6 the mean CRPSS reached values even above 0.2.

Regarding my minor comment on the AO index - what I wanted included in the paper was an explanation of how the AO index is calculated i.e. what procedure is applied to the 1000hPa geopotential height field to produce the AO index?

We have now added the explanation to the text:

[revised manuscript text omitted]

Weeks 1–2   Weeks 3–4   Weeks 5–6

ERA-Interim OBSERVATIONS 1981-2016
 after $SWI_{neg}$
 after $SWI_{plain}$

ERA-Interim OBSERVATIONS 1997-2016
 after $SWI_{neg}$
 after $SWI_{plain}$

ECMWF REFORECASTS 1997-2016
 after $SWI_{neg}$
 after $SWI_{plain}$

Mean surface temperature anomaly (°C)

**Figure 46. ERA-Interim observed (a-l) and ECMWF reforecasted (m-r) mean temperature anomalies in comparison to the 1981-2016 mean during boreal winters (November-February) in cases the previous week's $SWI$ was negative ($SWI_{neg}$, covering about 21% of the winter weeks) or plain ($SWI_{plain}$, covering about 59% of the winter weeks). The dotted areas represent the 95% level of confidence where the means of surface temperature anomalies after $SWI_{neg}$ and $SWI_{plain}$ differ significantly.**

[Figure]

**Figure 57.** Sensitivity of the expected CRPSS of the post-processed ECMWF surface temperature reforecasts to the $k_{SWI}$ ranging from 0.0 to 1.0 in forecast weeks 3–4 (a) and 5–6 (b). The black boxes show the lower and upper quartiles, and the whiskers illustrate the extremes of the November-February mean CRPSSs of all the grid points in Northern Europe.

FORECAST WEEKS 3-4    FORECAST WEEKS 5-6

[Figure]

Mean bias-
corrected
ECMWF
reforecast

Mean bias-
corrected and
*SWI* post-
processed
ECMWF
reforecast

CRPSS

1.0

0.8

0.6

0.4

0.2

0.0

[Figure]

Figure 68. Expected CRPSS of forecast weeks 3–4 and 5–6 of the ECMWF's mean temperature reforecasts for November–February 1997–2016 after mean bias-correction (a-fb) and after both mean bias-correction and the *SWI* based post-processing (ge-hd). ERA-Interim climatology of 1981–2010 was used as the reference. The dotted areas represent the 95% level of confidence that the CRPSS is above zero.

---

## Author Response (AR3)

**Authors Response to comments on "Adding value to Extended-range Forecasts in Northern Europe by Statistical Post-processing Using Stratospheric Observations" by Natalia Korhonen et al.**

The comments are in Black and the responses in blue.

We thank the editor and the reviewer for their comments on the revised manuscript.

We have now done further changes to the manuscript. The *SWI* categories have now been defined using the ZMZW only and the forecast skill scores have been recalculated according to this. In addition to these, we have done further editing to the manuscript, to clarify it, according to the comments. Below we respond to the reviewer point-by-point.

Best regards

Natalia Korhonen

Submitted on 16 Apr 2020
Anonymous Referee #3

Having read the latest revision of the paper, I still don't feel that my main reservation about the analysis has been adequately addressed, and I therefore do not feel the paper is suitable for publication in its current form. In my previous reviews, I expressed scepticism about the use of the QBO as a predictor of the Arctic Oscillation (AO). It seemed more plausible to me that the polar vortex strength was a stronger predictor of the AO, and that the authors should more fully analyse this before focusing on the QBO.

In the latest draft the authors have revised their methods, but I now find the latest analysis rather confusing and hard to follow. If I understand correctly, the authors now use the polar vortex strength (ZMZW) as a predictor of the AO if the vortex is weak (ZMZW below its 10$^{th}$ percentile) or strong (ZMZW above its 80th percentile) and demonstrate that the AO and temperature are influenced by the polar vortex strength. For post-processing of temperature forecasts, however, they only consider cases where ZMZW is between the 10th and 80th percentiles, and use the QBO to categorise different forecast ensemble members, with a selection of different QBO wind thresholds tried. This choice of method appears arbitrary and is not adequately justified in the paper. It is hard to see a chain of reasoning that led to the final method, which seems rather complicated. Why restrict the post-processing to those cases where the ZMZW is between the 10th and 80th percentiles, when you've just shown that a weak or strong polar vortex has an effect on the AO and temperature?

The authors argue that the QBO is important, since they have left out the weak and strong ZMZW cases from the post-processing, and still find improved forecast skill when using the

QBO only to define their SWI_neg and SWI_plain categories. But this misses the point, in my opinion - if the QBO is influencing the AO via the polar vortex, then ZMZW is the only predictor needed. The SWI categories should therefore first be defined using ZMZW only, and the forecast skill calculated. Including the QBO in the SWI categories' definition is only then justified if it leads to a significant improvement in skill compared to using ZMZW only.

Response: As suggested in the third paragraph of the comment above, the SWI categories have now been defined using the ZMZW only and the forecast skill calculated. As the forecast skill was similar (i.e., not significant difference) to the SWI including QBO, the QBO was no longer included in the SWI. We have updated Figures 2-5 and edited the text according to this modification, i.e., QBO is left out elsewhere but the Discussion part.

Minor comments

p6 line 15: Where did the choice of QBO wind threshold values come from? (7, 10, 13 ms-1)

Response: this part was now left out.

Figure 2 : the caption states that the p-values relate to pairs of distributions, but (unlike figure 3) it's not clear what these pairs are. Also, why are there only 57 members above the 80th percentile, but 149 members below the 20th percentile?

Response: The Fig. 2 has now been replotted, and the pairs of distributions are now visible.

The quantiles in the previous version were calculated directly from the November-March daily data. As the members represented 10 days minima, there were more members below the $20^{th}$ percentile than above the $80^{th}$ percentile. In the current revised manuscript the quantiles are now defined directly from the 10 days minima of the November-February ZMZW at 60N 10 hPa.

Figure 5,6 : The numerical values in the colour bar don't correspond to the boundaries between colours.

Response: the figures have now been replotted and now the numerical values in the color bar do correspond to boundaries between the colors.

[revised manuscript text omitted]

FORECAST WEEKS 3-4          FORECAST WEEKS 5-6

a)    b)

Mean bias-corrected
ECMWF reforecasts in
cases ZMZW≤3.8m/s at the
start of the forecast

c)    d)

Mean bias-corrected
ECMWF reforecasts in
cases ZMZW>41m/s
at the start of the forecast

CRPSS
- 1.0
- 0.8
- 0.6

e)    f)

Mean bias-corrected
ECMWF reforecasts in
cases SWI$_{neg}$/SWI$_{plain}$ at the
start of the forecast

- 0.4
- 0.2
- 0.0

g)    h)

Mean bias-corrected **and**
*SWI* **post-processed**
ECMWF reforecasts in
cases SWI$_{neg}$/SWI$_{plain}$ at the
start of the forecast

**Figure 8. Expected CRPSS of forecast weeks 3–4 and 5–6 of the ECMWF's mean temperature reforecasts for November–February 1997–2016 after mean bias-correction (a-f) and after both mean bias-correction and the *SWI* based post-processing (g-h). ERA-Interim climatology of 1981–2010 was used as the reference. The dotted areas represent the 95% level of confidence that the CRPSS is above zero.**